# Microengineered human blood–brain barrier platform for understanding nanoparticle transport mechanisms

Song Ih Ahn [1,2], Yoshitaka J. Sei[1,2], Hyun-Ji Park [1], Jinhwan Kim[1], Yujung Ryu[1], Jeongmoon J. Choi[3], Hak-Joon Sung [4], Tobey J. MacDonald[5], Allan I. Levey[6] & YongTae Kim [1,2,7,8]*

Challenges in drug development of neurological diseases remain mainly ascribed to the blood–brain barrier (BBB). Despite the valuable contribution of animal models to drug discovery, it remains difficult to conduct mechanistic studies on the barrier function and interactions with drugs at molecular and cellular levels. Here we present a microphysiological platform that recapitulates the key structure and function of the human BBB and enables 3D mapping of nanoparticle distributions in the vascular and perivascular regions. We demonstrate on-chip mimicry of the BBB structure and function by cellular interactions, key gene expressions, low permeability, and 3D astrocytic network with reduced reactive gliosis and polarized aquaporin-4 (AQP4) distribution. Moreover, our model precisely captures 3D nanoparticle distributions at cellular levels and demonstrates the distinct cellular uptakes and BBB penetrations through receptor-mediated transcytosis. Our BBB platform may present a complementary in vitro model to animal models for prescreening drug candidates for the treatment of neurological diseases.

[1] George W. Woodruff School of Mechanical Engineering, Georgia Institute of Technology, Atlanta, GA 30332, USA. [2] Parker H. Petit Institute for Bioengineering and Bioscience, Georgia Institute of Technology, Atlanta, GA 30332, USA. [3] School of Biological Sciences, Georgia Institute of Technology, Atlanta, GA 30332, USA. [4] Department of Medical Engineering, Yonsei University College of Medicine, Seoul 03722, Republic of Korea. [5] Department of Pediatrics, Emory University, Atlanta, GA 30322, USA. [6] Department of Neurology, Emory University, Atlanta, GA 30322, USA. [7] Wallace H. Coulter Department of Biomedical Engineering, Georgia Institute of Technology, Atlanta, GA 30332, USA. [8] Institute for Electronics and Nanotechnology, Georgia Institute of Technology, Atlanta, GA 30332, USA. *email: ytkim@gatech.edu

The blood–brain barrier (BBB) is a highly functionalized vascular border of the central nervous system (CNS) that regulates the transport of substances between the blood and brain[1]. The barrier function is attributed mainly to the unique perivascular structure specialized by a three-dimensional (3D) network of astrocytes that communicate with endothelial cells and pericytes (Fig. 1a)[2]. Astrocytes form the glia limitans of the BBB with their end-feet contacting the blood vessels and control the influx of water through aquaporin-4 (AQP4)[3,4]. Pericytes embedded in the basement membrane wrap around the endothelium and contribute to astrocytic polarization[5]. These complex cellular interactions at the BBB maintain its integrity and restrict the penetration of drugs, leading to a low success rate in the development of therapeutics for CNS diseases[6].

To deliver drugs across this barrier, CNS delivery systems have been widely explored to cross the BBB[7], including nanoparticle (NP)-mediated drug delivery with ligands specific to BBB endothelial surface receptors[8,9]. In particular, high-density lipoprotein (HDL)-mimetic NPs have been introduced as promising CNS delivery systems due to the innate endogenous character to facilitate the delivery of therapeutic molecules across the BBB via lipoprotein receptor-mediated transcytosis[10–12]. However, the lack of experimental models that can precisely evaluate the interactions between the BBB and delivery carriers restricts successful clinical translation of therapeutic and diagnostic NPs[13,14]. Animal models often do not predict drug responses in humans due to species differences[15–17]. Moreover, the complex physiology of animal models makes it difficult to perform mechanistic studies and direct quantitative analysis of NPs with the barrier at molecular and cellular levels in real time[13]. These challenges highlight the importance of developing an in vitro model that mimics the essential physiological structure and function of the human BBB and that reproduces the key relationships of healthy and disrupted barrier functions in a controlled manner.

Recent advances in organ-on-a-chip technology have provided the ability to recapitulate the microenvironment of the BBB[18,19]. Existing in vitro human BBB-on-chip models have made efforts to reconstitute the tight endothelial barrier function using several

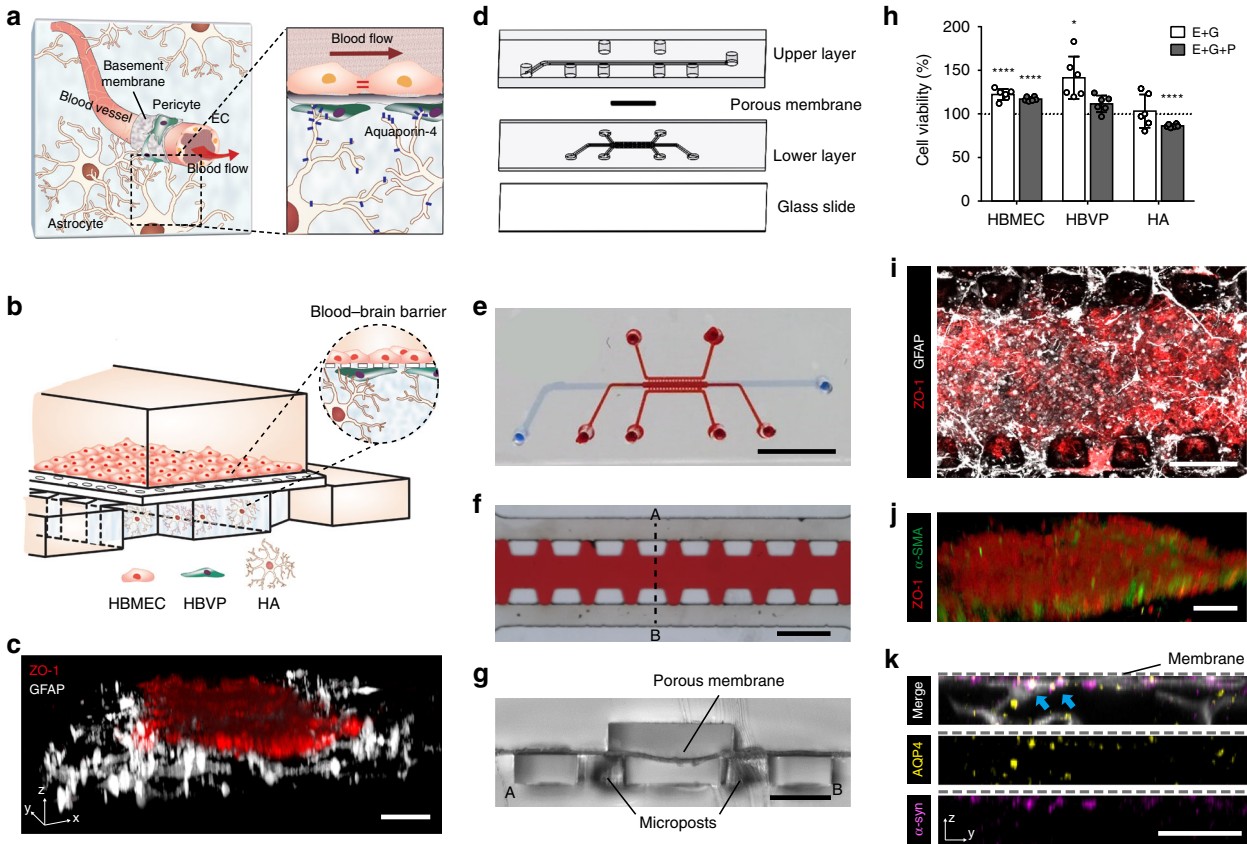

**Fig. 1 Microengineered human blood–brain barrier (BBB) model. a** Schematic description of the BBB consisting of endothelial cells (ECs) along the blood vessel under continuous blood flow, pericytes covering the endothelial monolayer, and astrocytes with aquaporin-4 (AQP4) expression at their end-feet near the blood vessel. **b** Schematic description of our microengineered human BBB model. **c** 3D configuration of the BBB model showing human brain microvascular endothelial cells (HBMECs) (ZO-1, red) and human astrocytes (HAs) (GFAP, white) (scale bar = 100 μm). **d** Explosion view of the device consisting of upper vascular layer, porous membrane, lower perivascular layer, and glass slide. **e** A photo of the device after completing fabrication of the device (blue: upper channel and red: lower channels) (scale bar = 500 μm). **f** Lower layer consisting of three parallel channels separated by series of micropillars (red: center channel) (scale bar = 500 μm). **g** Cross-section of the device after fabrication (along A-B in Fig. 1f) (scale bar = 200 μm). **h** Cell metabolic activities assessed by a (3-(4,5-dimethylthiazol-2-yl)-5-(3-carboxymethoxyphenyl)-2-(4-sulfophenyl)-2H-tetrazolium) (MTS) assay (E + G: 1:1:1 mixture of endothelial medium, astrocyte medium, and microglia medium, E + G + P: 1:1:1:1 mixture of endothelial medium, astrocyte medium, microglia medium, and pericyte medium) (Data represent mean ± s.d. of $n = 6$ for each condition, $*p < 0.05$ and $****p < 0.001$ versus each cell culture medium by student $t$-test). **i** Bottom view of the device with endothelial monolayer (ZO-1, red) and astrocytic network (GFAP, white) (scale bar = 50 μm). **j** Endothelial monolayer (ZO-1, red) supported by a layer of human brain vascular pericytes (HBVPs) (α-SMA, green) (scale bar = 50 μm). **k** Aquaporin-4 (AQP4, yellow) and α-syntrophin (α-syn, magenta) expressions at astrocytic end-feet (GFAP, white) underneath a porous membrane (indicated as the dotted line) in the lower channel (Blue arrows indicate co-localization of AQP4 with α-syn.) (scale bar = 50 μm). All images are representative ones from at least five biological and three technical replicates.

platforms that include monoculture of brain endothelial cells[20] and co-culture of endothelial cells with astrocytes in two-dimensional (2D)[21] and 3D[22,23] microenvironments. A recent BBB model with 3D culture of endothelial cells, pericytes, and astrocytes enabled reconstitution of direct cellular interactions, resulting in the barrier function with permeability lower than previous in vitro models of mono-culture or co-culture[24]. However, it remains difficult to incorporate the complex physiology of astrocytes into the in vitro BBB models and demonstrate the contribution of astrocytes to the BBB's health and disruption. In a healthy brain, astrocytes show branched processes around their cell bodies[25] and concentrated expression of AQP4 at their end-feet processes near vasculature[4,26,27]. In response to CNS injuries or diseases, astrocytes become reactive, which contribute to BBB breakdown and disease progression such as neurodegeneration, ischemia, and infection[28]. Reactive astrocytes undergo changes in morphology and gene expressions including lipocalin-2 (LCN2) and Serpin Family A Member 3 (Serpina3)[29]. In vitro BBB models thus would be enhanced and widely useful for CNS disease modeling if healthy and reactive astrocytes could be reconstituted in a controlled manner.

Here we present a microengineered physiological system designed to model the human BBB with brain endothelial cells, pericytes, and 3D astrocytic network (Fig. 1b). The use of human brain microvascular endothelial cells (HBMECs) in our BBB chip enables the reproduction of BBB-specific endothelial characteristics with greater gene expressions of junctional markers, membrane transporters and receptors, resulting in a tight barrier with permeability comparable to recent tri-culture models. Importantly, our BBB chip creates a 3D astrocytic network with polarized expression of AQP4 and decreased reactive gliosis markers including LCN2. The reduced reactive astrogliosis allows for neuroinflammation modeling in response to extrinsic stimuli of pro-inflammatory factors such as interleukin-1β (IL-1β). The model combined with high-precision sampling and flow cytometric analysis enables on-chip quantification of nanoparticle transport and distribution in the vascular and perivascular regions at cellular and tissue levels. Our BBB chip may provide a reliable tool for better understanding of drug distribution and efficacy at the BBB in both physiological and pathological conditions.

## Results

**Microengineered human BBB model with 3D astrocytic network**. Our microengineered human BBB model reconstitutes the BBB structure with the brain vascular endothelium and human brain vascular pericytes (HBVPs) in direct contact with 3D network of human astrocytes (HAs) (Fig. 1b, c). The BBB chip has two compartmentalized microfluidic channel layers that combine a 2D endothelial monolayer with a 3D brain microenvironment, enabling highly sensitive quantification of molecular distribution in each space independently (Fig. 1d, e). The upper layer of the device mimics the vascular space of the brain microvasculature where an endothelial monolayer is formed on a 7 μm thick porous membrane (8 μm diameter pores at a density of 1E5 pores cm$^{-2}$) with 16 μL min$^{-1}$ of continuous fluid flow (shear stress: 4 dyne cm$^{-2}$). The lower layer accommodates pericytes underneath the membrane and astrocytes in a 3D Matrigel (5 mg mL$^{-1}$) in the center channel along with the two side channels (Fig. 1f). This structure allows for the 3D astrocyte culture in a hydrogel that is inserted into the center channel and is stably maintained by surface tension. Importantly, the edges of the upper channel are aligned to cover the both side arrays of micropillars (Fig. 1g) to avoid the undesirable leakage that may occur in the edge of a microfluidic device of rectangular cross-section due to

heterogeneous formation of an endothelial monolayer at the channel walls, which we recently demonstrated with a microfluidic transcellular monitor[30]. The device is designed to have diffusive transport of culture medium components into the hydrogel channel with media refreshment in the upper and the two side channels (Supplementary Fig. 1). Moreover, the two side channels promote independent lateral perfusion through the hydrogel enabling efficient removal of metabolic wastes or unbound antibodies and, more importantly, provide the opportunity to precisely sample solutions perfused from the hydrogel without disturbing the cellular organization.

With morphological and metabolic activity assays (see Methods for details) in the co-culture medium selection (Fig. 1h and Supplementary Fig. 2), we developed a culture protocol (Supplementary Figs. 3 and 4), with which we were able to establish the human BBB integrity after 2.5 days of culture, as the culture time was also suggested and validated in previous studies[21,31]. Our BBB model reproducibly maintains the 3D network of HAs with physiologically relevant morphology underneath the endothelial monolayer and across the lower channel layer (Fig. 1i). The monolayers of HBMECs and HBVPs on the opposite sides of the 7 μm-thick porous membrane (Fig. 1j) while the proximity and perfusable structure allow for the paracrine and juxtacrine signaling between the three cells. More importantly, our BBB chip clearly demonstrated that HAs in a 3D network extend their end-feet with higher AQP4 and α-syntrophin (α-syn) expressions right underneath the porous membrane over the basal side of the endothelium (Fig. 1k).

**Brain-specific endothelial barrier fuction**. The brain vascular endothelium is a highly specialized gatekeeper with complex transport mechanisms[6,32]. The critical barrier function of the BBB endothelium is reportedly characterized by high expressions of BBB-specific proteins[33,34], including junctional, transporter, and receptor proteins and by high transendothelial electrical resistance (TEER) (i.e., low permeability)[35]. In addition, applications of specific human cell sources have enhanced the physiological relevance of in vitro models to the unique properties of the BBB endothelium[24]. In our present study thus, we first demonstrated that our brain-specific endothelial cells, when cultured with the other BBB cells (i.e., astrocytes and pericytes), exhibited increases in representative gene expressions including proteins that regulate junctional formation, carrier-mediated transport, active efflux, and amyloid beta (Aβ) transport (Fig. 2a). Especially, BBB cellular interactions upregulated the endothelial gene expressions of the representative junctional proteins such as occludin (OCLN), zonula occludens-1 (ZO-1), and vascular endothelial cadherin (VE-cad) (Fig. 2b) and the representative membrane transporters and receptors including glucose transporter 1 (GLUT1), cholesterol efflux regulatory protein (CERP; ATP-binding cassette subfamily A member 1, ABCA1), and low-density lipoproetin receptor-related protein 1 (LRP1) (Fig. 2c).

In our BBB chip, HBMECs cultured in the vascular channel under a physiological shear stress (4 dyne cm$^{-2}$)[36] establish an intact monolayer with tight junctions (Fig. 2d) while HBVPs are cultured on the other side of the porous membrane in the perivascular channel (Fig. 2e). More importantly, HAs in a hydrogel of the perivascular channel exhibit a typical star-shaped morphology with radial distribution of fine branches and their 3D cellular network (Fig. 2f, g). This 3D cellular structure in our BBB chip allows for the highly complex yet organized BBB construction (Fig. 2h–j), reproducibly leading to the greater barrier function (Fig. 2k). A physiological level of shear stress was responsible for inducing endothelial function with the barrier tightness (Fig. 2l), efflux transporter protein expression

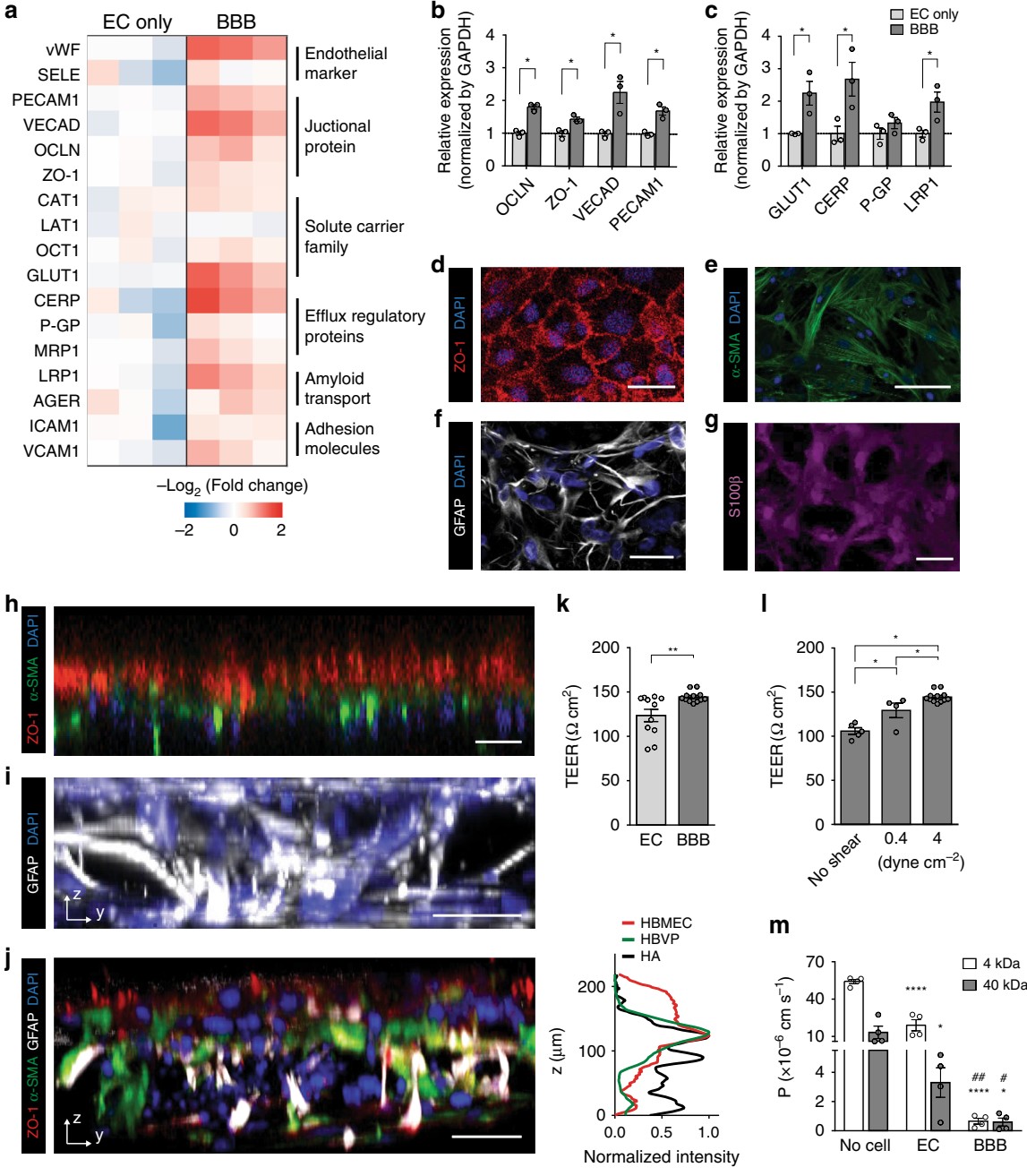

**Fig. 2 Barrier integrity of the endothelial monolayer for the BBB chip. a** Heat map of RT-qPCR results of HBMECs in mono-culture and tri-culture systems ($n = 3$ for each condition). **b, c** Gene expression of HBMECs in mono-culture and tri-culture systems including junctional proteins (**b**) and receptor proteins (**c**) ($n = 3$ for each condition, *$p < 0.05$ by student $t$-test). **d** Tight endothelial monolayer (ZO-1, red; DAPI, blue) formed in the upper channel of the device. **e** Pericytes cultured underneath the porous membrane where an endothelial monolayer is constructed on the other side (α-SMA, green; DAPI, blue). **f, g** Astrocytes with star-shaped morphology labeled with GFAP (GFAP, white) (**f**) and S100β (S100β, magenta) (**g**). **h** Bi-layer of an endothelial monolayer (ZO-1, red) and HBVPs (α-SMA, green). **i** Astrocytic end-feet stretching to the endothelium in 3D cellular network (GFAP, white; DAPI, blue). **j** 3D BBB structure constructed in a device (ZO-1, red; α-SMA, green; GFAP, white; DAPI, blue). The fluorescence intensity profiles indicate the distribution of ZO-1, α-SMA, and GFAP in the image. **k** Transendothelial electrical resistance (TEER) measured across the membrane between the upper and lower layers with an endothelial monolayer (EC) and an endothelial monolayer with pericytes and astrocytes (BBB) ($n = 11$ for EC and $n = 12$ for BBB, **$p < 0.01$ by student $t$-test). **l** TEER measured from BBB models under different levels of shear stress ($n = 5$ for No shear, $n = 4$ for 0.4 dyne cm$^{-1}$, and $n = 12$ for 4 dyne cm$^{-2}$, *$p < 0.05$ by student t-test). **m** Permeability coefficients calculated from the diffusion of 4 kDa and 40 kDa FITC-dextran through a membrane (No cell), an endothelial monolayer (EC), an endothelial monolayer co-cultured with pericytes and astrocytes (BBB) ($n = 4$ for each condition, *$p < 0.05$ and ****$p < 0.001$ vs. No cell, #$p < 0.05$ and ##$p < 0.01$ vs. EC, all by student $t$-test). Data are presented as mean ± s.e.m. All scale bars = 50 μm. All images are representative ones from at least five biological and three technical replicates.

(Supplementary Fig. 5), and endothelial nitric oxide synthase (eNOS) phosphorylation (Supplementary Fig. 6a, b). Moreover, our model showed size-dependent molecular transport across the BBB (Fig. 2m). The TEER values and permeability coefficients to 4 kDa and 40 kDa FITC-dextran were comparable to previous BBB studies[21,24,31]. The permeability coefficient of our BBB model reached as low as the values measured in vivo[37].

**Reduced reactive gliosis of 3D-cultured astrocytes.** Not only is the vascular endothelial barrier function important to developing and validating in vitro BBB models, the physiological relevance provided by perivascular regions is also essential to precisely recapitulate the BBB structure and function. One difficult yet important element in reconstituting the BBB is to preserve the morphological and physiological characteristics of healthy astrocytes[38]. Astrocytes reportedly restore their in vivo-like physiological properties such as morphology and functional reactivity in 3D culture systems[25,39]. We confirmed that HAs cultured on a 2D Matrigel-coated surface were flat and polygonal in shape (Fig. 3a and Supplementary Fig. 7a). However, HAs cultured in 3D Matrigel exhibited more in vivo-like ramified morphology (Fig. 3b and Supplementary Fig. 7b). Moreover, the majority of HAs cultured in 3D featured small cell bodies with radially distributed thin and long branches, whereas HAs cultured in 2D exhibited enlarged cell bodies with less and short processes (Fig. 3c, d).

Reactive astrocytes are characterized by changes in gene expression as well as morphology. We performed quantitative analysis on the gene expression of reactive gliosis markers that are upregulated in pathological conditions to support that 3D-cultured HAs in our BBB chip are more physiologically relevant than conventional 2D culture systems. In particular, LCN2 plays an important role in neuroinflammation by mediating pro-inflammatory responses in injury[40]. We found that reactive gliosis markers, vimentin (VIM) and LCN2, were downregulated in HAs cultured in 3D compared to those cultured in 2D, while the level of glial fibrillary acidic protein (GFAP), the representative astrocyte marker, did not significantly change (Fig. 3e). In addition, we confirmed that the LCN2 expression level could be further regulated in a dose-dependent manner in response to an inflammatory cytokine treatment with IL-1β in the 3D culture as in 2D (Fig. 3f, g). These results demonstrate the physiological relevance of 3D-cultured HAs in our BBB chip and indicate the potential application for reactive astrogliosis modeling such as in neuroinflammation.

**Polarized expression of aquaporin-4 in the BBB chip.** Astrocytes in the perivascular space play a role in regulating the water homeostasis in the brain via the water channel protein AQP4 at their end-feet processes[26,41]. Localization of AQP4 in astrocytic end-feet processes is therefore important to recapitulate the BBB in homeostatic and physiological conditions. Our BBB model recapitulates a complex 3D network of HAs with expression of AQP4 along their branches while 2D culture models show diffusively expressed AQP4 in plasma membrane of astrocytes (Supplementary Fig. 8). We analyzed AQP4 polarization by calculating the ratio of AQP4 labeled along astrocytic end-feet in a vascular side versus that in a parenchymal side of the perivascular channel as previously reported[4] (Fig. 3h) and by demonstrating co-localization with α-syn, the immediate anchor for AQP4 that controls AQP4 polarization to astrocytic end-feet, in the model (Fig. 3i). The localization of AQP4 in HAs was polarized to the astrocytic end-feet in the vascular side of the channel when cultured with HBMECs and HBVPs (Fig. 3j, k). The polarized distribution of AQP4 was significantly induced in the presence of

HBVPs, as previously observed in vivo[5]. This finding implies that our model can mimic the water transport system at the BBB, which is responsible for the homeostasis of ions and water in the brain.

**Nanoparticle transport analysis on chip.** One challenge in CNS drug delivery research is to accurately quantify the transport of drug compounds into the brain at cellular and molecular levels. Our BBB chip allows for the monitoring of the interactions between cells and NPs but also enables us to quantify the distribution of NPs in vascular and perivascular spaces as well as in each cell type. To demonstrate these abilities of our system, we first synthesized a bioinspired nanoparticle that mimics HDL. We used our microfluidic technology to engineer HDL-mimetic nanoparticles with apolipoprotein A1 (eHNP-A1) (Fig. 4a), which reconstitute the physiologically relevant size and composition to discoidal HDL (Fig. 4b–d). We confirmed through a biodistribution study following intravenous injection of the fluorescent-dye labeled eHNP-A1 that eHNP-A1 can enter the BBB with ~3% of relative accumulation in the brain (Fig. 4e, f and Supplementary Fig. 9). Confocal imaging analysis of the cryosectioned brain tissue confirmed that the systemically administered eHNP-A1 were localized around the cell nucleus (Fig. 4g).

We then quantified eHNP-A1 distributions in the vascular and perivascular spaces at the BBB and studied the transport mechanisms in a controlled manner. We hypothesized that when a NP solution is introduced into our BBB chip, NPs either remain in the vascular channel, interact with HBMECs, or translocate into the perivascular channel in which NPs can interact with HBVPs or HAs (Fig. 4h–j). With the results of eHNP-A1's BBB penetration as shown in our animal study (Fig. 4e–g), we used our BBB chip to investigate the mechanism by which eHNP-A1 can get into the brain. We tested the hypothesis that eHNP-A1 primarily leverages scavenger receptor class B type 1 (SR-B1) on brain endothelial cells to cross the BBB via transcytosis, one of the major transport mechanisms of natural HDL[42,43]. After blocking SR-B1, the amount of eHNP-A1 remaining in the vascular channel significantly increased (Fig. 4k), whereas the amount of eHNP-A1 translocated to the perivascular channel was not significantly changed (Fig. 4l). This result indicates that blocking SR-B1 activity reduces eHNP-A1 uptake by HBMECs, whereas it may induce a compensatory upregulation of other receptors or determinants of endothelial transcytosis such as SNAREs[42,44]. We then mapped the 3D distribution of eHNP-A1 in our BBB chip by compensating NP loss caused by the adsorption to the PDMS surface (Supplementary Fig. 10). Our results revealed that block lipid transporter-1 (BLT-1) treatment reduced the amount of eHNP-A1 that gets into the tissue (EC uptake and perivascular channel) by ~3 fold (Fig. 4m, n). Cellular uptake of eHNP-A1 was further quantified using fluorescence-activated cell sorting (FACS) analysis. The total portion of eHNP-A1 positive cells was decreased in the BLT-1 treated BBB models (Fig. 4o, p), indicating that SR-B1 mediates the cellular uptake of eHNP-A1 in our BBB chip as shown in previous study[42]. In particular, the portion of perivascular cells (i.e., HBVPs and HAs) that have taken up eHNP-A1 was decreased by 38.5% and 53.8%, respectively with BLT-1 treatment, whereas the portion of HBMECs with eHNP-A1 was not changed (Fig. 4q).

**Discussion**

Our on-chip human BBB model successfully recapitulated the key structure and function of the BBB featuring the highly specialized brain endothelial monolayer and physiological network of astrocytes. The hybrid design combining two vertical layers with three parallel channels enabled the complex BBB co-culture, not

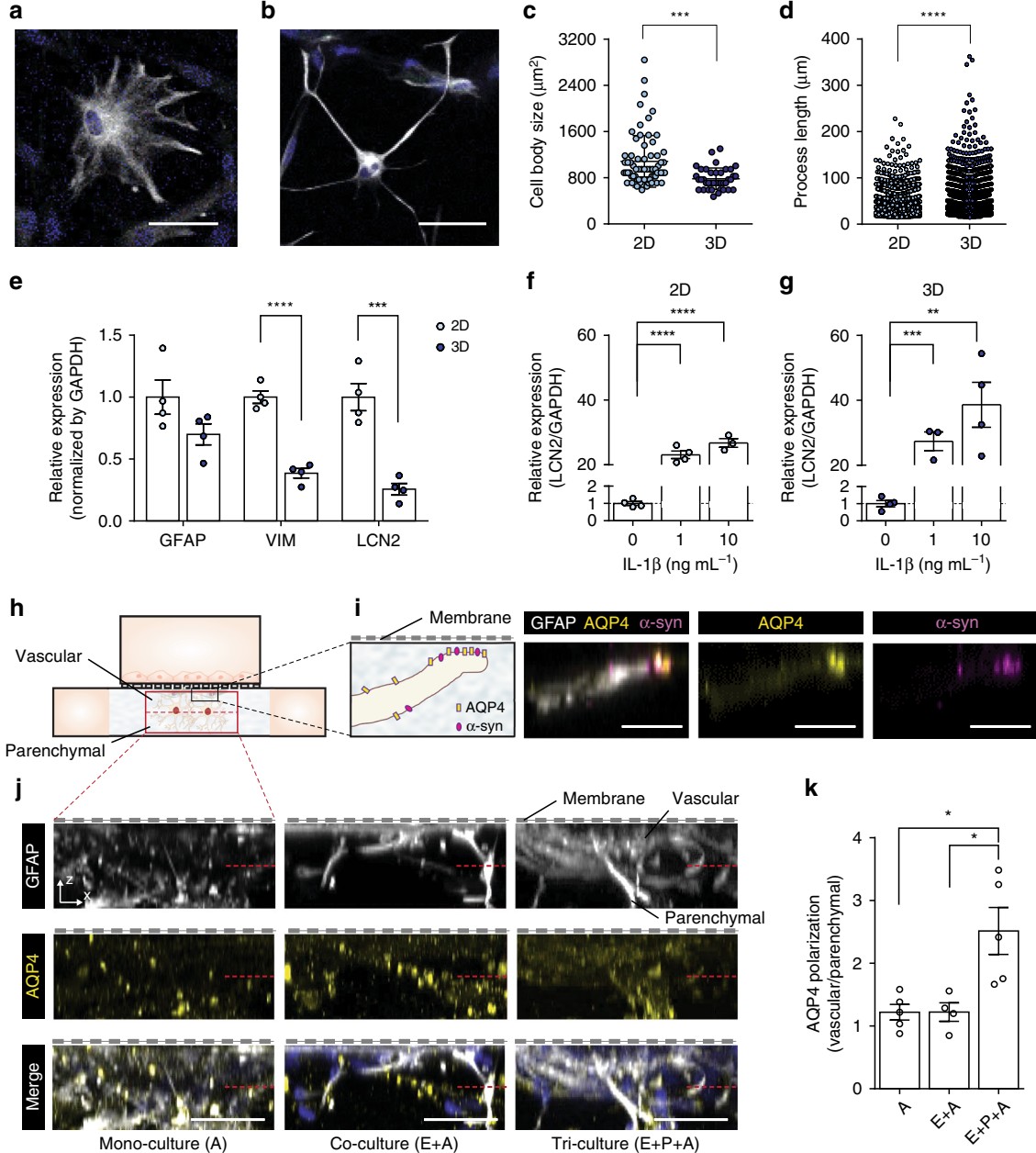

**Fig. 3 3D culture of astrocytes with AQP4 polarization in the BBB chip. a**, **b** Representative morphology of human astrocyte (HA) cultured on a Matrigel-coated 2D surface (**a**) and in 3D Matrigel (**b**) (GFAP, white; DAPI, blue) (scale bars = 50 μm). **c** Cell body size of HAs cultured in 2D and 3D ($n = 69$ for 2D and 39 for 3D, ***$p < 0.005$ by student $t$-test). **d** Process length of HAs cultured in 2D and 3D ($n = 1352$ for 2D and 1302 for 3D, ****$p < 0.001$ by student $t$-test). **e** Gene expression of reactive gliosis markers in HAs cultured in 2D and 3D ($n = 4$ for each condition, ***$p < 0.005$ and ****$p < 0.001$ by student $t$-test). **f**, **g** Gene expression of lipocalin-2 (LCN2) in HAs cultured on a Matrigel-coated 2D surface (**f**) and within 3D Matrigel (**g**) with interleukin-1β (IL-1β) treatment demonstrating the ability to model reactive astrocytes more effectively in 3D ($n = 4$ for each condition, **$p < 0.01$, ***$p < 0.005$, and ****$p < 0.001$ by student $t$-test). **h** Quantitative analysis of AQP4 polarization by measuring AQP4 distribution in vascular and parenchymal side in the perivascular channel. **i** Co-localization of AQP4 (AQP4, yellow) and α-syn (α-syn, magenta) at astrocytic end-feet (scale bars = 20 μm). **j** Distribution of AQP4 (AQP4, yellow) along the cell bodies of HAs (GFAP, white; DAPI, blue) in the channel (scale bars = 50 μm). **k** Polarized expression of AQP4 to the vascular side in the perivascular channel ($n = 4$ for each condition, *$p < 0.05$ by student $t$-test). Data are presented as mean ± s.e.m. All images are representative ones from at least five biological and three technical replicates.

only establishing a 2D intact brain endothelium and reconstructing a 3D brain microenvironment with astrocytic network but also connecting the BBB cells in perfusable proximity for their intercellular signaling in co-culture. Moreover, the side channels along with the 3D brain microenvironment allowed for a precise quantification of NP distributions across the BBB.

Recent in vitro models have demonstrated the importance of cell source to mimic organ-specific fuction in vitro, as well as for

human disease modeling[45,46]. In particular, highly complex BBB organization requires brain-specific cells to be sourced appropriately for in vitro modeling, which secures the key characteristics including the tight barrier function and low permeability resulting from high expressions of BBB-specific proteins[6,32–34], compared to human umbilical vein endothelial cells (HUVECs) widely used in previous models[23,47] that may not closely recapitulate the unique properties of the brain endothelium[48].

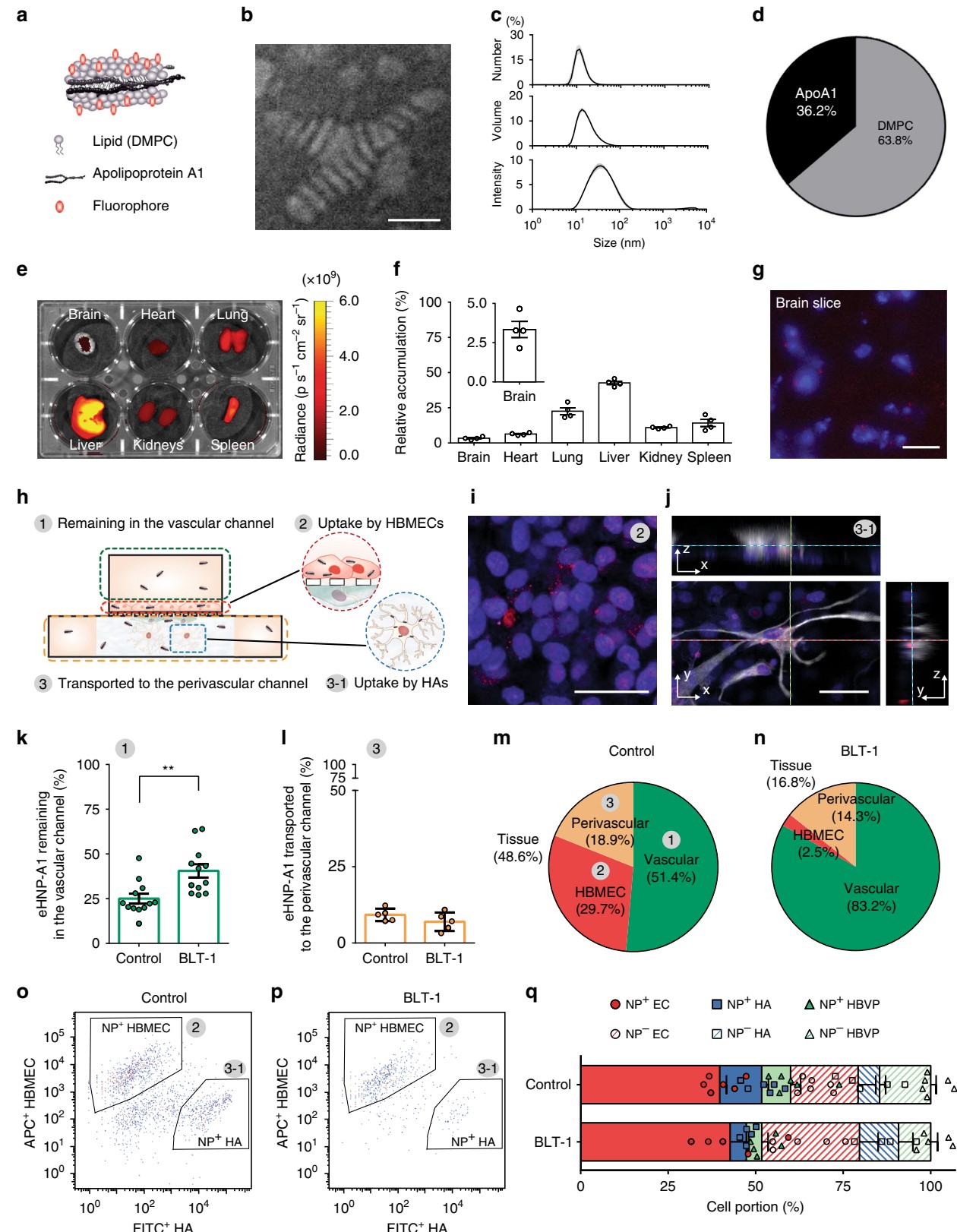

Immortalized human brain endothelial cells are good candidates for standardized screenings due to their availability as well as indefinite proliferation while preserving their properties. Previous studies using hCMEC/D3 in microfluidic BBB models showed their barrier function with junctional protein expression, TEER[49], and permeability[47]. In our present study, we have used HBMEC as it has been reported that HBMEC is the most suitable and promising immortalized human brain endothelial cell line among the four available cell lines (hCMEC/D3, HBMEC, TY10, and BB19) for in vitro modeling of human BBB in terms of the barrier tightness and permeability[50]. Moreover, the use of HBMECs validated their BBB-specific function with their increased gene

**Fig. 4 On-chip BBB transport analysis of HDL-mimetic nanoparticles. a** Discoidal engineered HDL-mimetic nanoparticle with apolipoprotein A1 (eHNP-A1) consisting of lipid, apolipoprotein A1 and fluorescent marker. **b** Transmission electron microscopy (TEM) image of the synthesized eHNP-A1 (scale bar = 20 nm). **c** Size distribution of the synthesized eHNP-A1. **d** Composition of the eHNP-A1. **e** Biodistribution of the eHNP-A1. **f** Quantification of the relative fluorescence intensity in each organ (Data represent mean ± s.e.m from n = 4 for each condition). **g** eHNP-A1 accumulated in the mouse brain (scale bar = 50 μm). **h** Schematic description of eHNP-A1 distribution in the BBB model showing (1) eHNP-A1 remaining in the vascular channel, (2) eHNP-A1 interact with endothelial cells (HBMECs), (3) eHNP-A1 translocated to the perivascular channel, and (3-1) eHNP-A1 interact with astrocytes (HAs). **i, j** Confocal images showing eHNP-A1 within the HBMEC monolayer (**i**) and HAs (**j**) in a BBB chip (scale bars = 50 μm). **k, l** Relative fluorescence intensity of sampled culture medium containing eHNP-A1 from the upper channel (**k**) and the lower channel (**l**) after 2 h of eHNP-A1 incubation in the vascular channel (**k**: n = 12; **l**: n = 5, **p < 0.01 by student t-test). Data show mean ± s.e.m. **m, n** Distribution of eHNP-A1 in control (**m**) and the block lipid transport-1 (BLT-1) treated microengineered BBB model (**n**). **o, p** Representative fluorescence-activated cell sorting (FACS) plot for the numbers of eHNP-A1 positive HBMECs and HAs in control (**o**) and BLT-1 treated BBB models (**p**). **q** Cellular uptake of eHNP-A1 in the BBB chip quantified from FACS analysis (n = 3). Data are presented as mean ± s.e.m. All images are representative ones from at least three biological and three technical replicates.

expressions of BBB-specific proteins in the presence of pericytes and astrocytes.

We have demonstrated the tight barrier integrity of our BBB model with the permeability coefficients comparable to those measured in vivo[37]. In addition to our evaluation of the barrier function with molecular permeability (permeability coefficient analysis), we measured the TEER values from the electrode wires inserted in the end of the inlet and outlet of the channels, as shown in previous studies[51–53], in order to cross-validate the barrier function difference between our models in the same device (not comparing them with Transwell or other models). We note that 3D astrocyte-laden hydrogel in the lower channel prevents electrodes from being coated along the bottom hydrogel channel as reported in recent studies[22,54]. We also note that the TEER values measured in our current study can be affected by the uneven distribution of the potential across the membrane and may not indicate the accurate values of the cell layer resistance as reported in recent studies[55–57]. The TEER values measured in this study thus were simply used to compare the barrier integrity between the EC only and the BBB models in the same device for the purpose of cross-validation of permeability coefficient analysis. For more information, we have included the raw resistance data in Supplementary Table 2.

Our BBB chip incorporated 3D culture of HAs into the cellular organization in order to recapitulate the BBB physiology and integrity. We demonstrated reduced astrogliosis in 3D-cultured HAs of an in vitro BBB model with the decreased expression of reactive gliosis markers as well as the in vivo-like morphology. A recent significant study has shown that LCN2 is a reactive astrocyte-specific marker that is commonly induced in pathological conditions, whereas intermediate filament proteins such as GFAP and VIM are normally expressed by astrocytes in both physiological and pathological conditions[29]. We demonstrated that LCN2 expression in HAs cultured in 3D was lower than in those cultured on 2D and more importantly that IL-1β treatment was more effective in 3D than in 2D, indicating the greater value for neuroinflammation modeling. Our results suggest that the cells restore their normal responses in 3D, while they are already reactive in 2D culture without external stimulation, revealing that studies that have used astrocytes in 2D culture may have overlooked the potential reactivity in their model. Moreover, the present BBB model shows the polarized distribution of AQP4 in perivascular astrocytes, which is critical to mimic the BBB physiology that maintains water and ion homeostasis in the brain. AQP4 is also involved in brain pathophysiology including glial scar formation[58] and neuroinflammation[59]. Taken together, our model established the physiologically relevant human BBB with the ability to control status of astrocytes from resting to reactive conditions as a potential for modeling of neuroinflammation and reactive gliosis in CNS diseases.

The brain-blood ratio of a drug concentration, a key parameter to estimate brain pharmacokinetics and brain-targeting efficiency,

has been previously assessed using rodents by in situ brain perfusion, brain microdialysis, and neuropharmacokinetic study[60,61]. Given the complexity of the techniques and species differences between humans and rodents, there is a critical need for the strategies that quickly and reliably measure how much drugs can penetrate to the brain parenchyma in a dose-dependent manner at multiple time points[62]. Our BBB model integrated with precise time-lapse sampling and end-point FACS analysis allowed us to precisely quantify 3D nanoparticle distribution at the BBB. Compartmentalized structure of the BBB chip conferred the ability to measure the amount of molecules separately for each space, allowing for quantitative assessment of BBB penetrations. Quantitative analysis of cellular uptakes in the BBB chip enabled us to evaluate the targeting efficacy of nanoparticles at cellular levels. Moreover, our model provided an in depth mechanistic understanding of the interactions between the BBB and nanoparticles at cellular levels.

To assess NP transport across the BBB, we used a HDL-mimetic NP synthesized with our microfluidic technology, which we recently reported[63]. Using our BBB chip, we demonstrated that eHNP-A1 is a potential CNS drug delivery system with their biomimetic ability to cross the BBB via SR-B1 mediated transcytosis. Our approach to 3D mapping of nanoparticle distributions in the vascular and perivascular spaces at the BBB will impact drug delivery and organs-on-chips areas.

In summary, we have presented a microengineered human BBB platform with physiologically relevant structure and function, offering its application in quantitative assessment of CNS drug delivery systems. Our BBB model allowed for multiple analyses including TEER measurement, nanoparticle sampling, and FACS analysis while the multiple devices were regulated by multi-syringe racks at the same time (Supplementary Fig. 4a). Microfluidic parallelization technology that integrates multiple devices while preserving advantages of the microscale organ-on-a-chip engineering will further provide higher throughput system[64]. We believe that our human BBB model could provide a widely useful tool for translational medicine research in particular for the modeling of neuroinflammation and reactive gliosis in neurological disorders.

## Methods

**Design and fabrication of the microfluidic device.** The microfluidic device was designed to have the widths of the upper, lower center, and lower side channels of 400 μm, 300 μm, and 200 μm, respectively. The height of all channels is 100 μm. The microfluidic device was fabricated with polydimethylsiloxane (PDMS; Sylgard 184; Dow Corning, Midland, MI, USA) using soft lithography[65]. To create the PDMS slab for the upper layer, PDMS pre-polymer (10:1 elastomer base to curing agent, wt/wt) was degassed and poured onto silicon wafers patterned with SU-8 (Microchem, Newton, MA, USA). The thin PDMS sheet for the lower layer of the device was created by spin coating an SU-8 patterned silicon wafer with a PDMS pre-polymer to a height of 250 μm. After curing the PDMS pre-polymer for 1 h at 80 °C, inlets and outlets of the channels were formed with a 1 mm diameter biopsy punch. A polycarbonate membrane (8 μm pore; Sterlitech Corp, Kent, WA, USA) treated with 5% 3-aminopropyl-trienthoxysilane (APTES) solution (Sigma-Aldrich,

St. Louis, MO, USA)[66] was sandwiched and bonded between the upper and lower PDMS layers using a plasma cleaner (Harrick Plasma, Ithaca, NY, USA). The fabricated device was then placed in a polystyrene box (Ted Pella, Redding, CA, USA). The device and microchannels are sterilized with 70% ethanol and placed in a dry oven at 80 °C for 2.5 days to restore hydrophobicity of the PDMS surface.

**Computational fluid dynamics**. Serum transport in the microfluidic channels was modeled with COMSOL (COMSOL, Multiphysics 5.3a, Stockholm, Sweden). The convection-diffusion equation was solved using the chemical species transport module. The diffusion coefficient of serum was assumed to be $5.9 \times 10^{-11}$ m$^2$ s$^{-1}$ in culture medium[67], and $8.0 \times 10^{-11}$ m$^2$ s$^{-1}$ in Matrigel[68]. The initial serum concentrations of culture medium and Matrigel was taken as $5 \times 10^{-3}$ mol m$^{-3}$ and 0 mol m$^{-3}$, respectively. Matrigel and the porous membrane were defined as a porous medium with porosity value of 0.8 and 0.016, respectively. The computational simulation was performed within a microfluidic device without cellular components. The three-dimensional model was used with no-slip boundary conditions on all the walls, and the numerical grid for performing the simulations consisted of ~850,000 finite elements.

**Cell culture**. Immortalized human brain microvascular endothelial cells (HBMEC; Innoprot, Bizkaia, Spain; #P10361-IM) at passage 5–10 were maintained in endothelial cell medium (Sciencell, San Diego, CA, USA) on flasks coated with 50 µg mL$^{-1}$ fibronectin (Sigma-Aldrich). Human brain vascular pericytes (HBVP; Sciencell; #1200) and human astrocytes (HA; Sciencell; #1800) were cultured on 1 mg mL$^{-1}$ poly-L-lysine (PLL, Sigma-Aldrich) coated flasks and maintained in astrocyte and pericyte medium, respectively (Sciencell). Both primary cells between passages 3 and 5 were used for all experiments. Fibronectin and PLL coating procedures were achieved following the manufacturer's instruction. To optimize cell culture condition, cell metabolic activity in different medium condition were quantified using a (3-(4,5-dimethylthiazol-2-yl)-5-(3-carboxymethoxyphenyl)-2-(4-sulfophenyl)-2H-tetrazolium) (MTS) assay ($n = 6$), which measures the formazan product by cell metabolic activity. CellTiter 96 AQueous One Solution Cell Proliferation Assay (Promega, Madison, WI, USA) was added to HBMECs, HBVPs, and HAs cultured in Endothelial cell medium (E), Astrocyte medium (A), Pericyte medium (P), Microglia medium (M; Sciencell), E + G (E:A:M = 1:1:1:1), and E + P + G (E:P:A:M = 1:1:1:1) after 3 days of culture. Here, we describe 1:1 mixture of A and M as glial cell medium (G). After 4 h of incubation, the optical absorbance of each sample was measured at 490 nm using Cytation 5 plate reader (BioTek, Winooski, VT, USA). To compare HAs in 2D and 3D, HAs were seeded at a density of $1 \times 10^6$ cells mL$^{-1}$ in Matrigel (Growth factor-reduced; Corning, NY, USA) coated 24 wells and in 3D Matrigel (5 mg mL$^{-1}$). After 1 day of culture, cells were stimulated with 1 ng mL$^{-1}$ and 10 ng mL$^{-1}$ of recombinant human interleukin-1-beta (IL-1β; Gibco, Grand Island, NY, USA) and incubated for another 20 h for analysis.

**Construction of the BBB chip system**. Prior to seeding HBVPs into the center channel of the lower layer, the channel was coated with 50 µg mL$^{-1}$ fibronectin (Sigma-Aldrich) for 1 h at 37 °C while the device was placed upside down. Then HBVPs were seeded at $1 \times 10^7$ cells mL$^{-1}$ density into the center channel and incubate for 6 h to allow adhesion of cells onto the fibronectin-coated poly-carbonate membrane. Then $1 \times 10^6$ HAs suspended in a 100 µL of Matrigel solution (Growth factor-reduced; Corning) were seeded into the same channel that HBVPs are cultured. The final concentration of Matrigel was calculated to be 5 mg mL$^{-1}$. After gelation of Matrigel in the channel by incubating at 37 °C for 30 min, cell culture medium was filled into the two side channels to avoid the gel drying out. The device was placed in the incubator at 37 °C for 6 h, and the upper luminal channel of the device was coated with 50 µg mL$^{-1}$ fibronectin for 1 h at 37 °C. HBMECs were then seeded into the upper channel with the density of $7 \times 10^7$ cells mL$^{-1}$. The final cell number ratio between HBMECs and HBVPs in a device was 1.5:1, and the ratio between HBMECs and HAs was 2:1, which was optimized for HAs to cover ~99% of the perivascular surface of the endothelium. After 24 h of culture to stabilize cells in the microfluidic device, the upper channel was connected to PhD Ultra syringe pumps (Harvard Apparatus, Holliston, MA, USA) and exposed to the media flow with the flow rate of 16 µL min$^{-1}$ to give cells the shear stress of 4 dyne cm$^{-2}$, which corresponds to the shear stress levels in the brain[36].

**Real-time qRT-PCR**. Cell-specific gene expressions were quantified by qRT-PCR analysis. Briefly, total RNA from HBMECs and HAs were isolated and collected using the RNeasy Mini kit (Qiagen GmBH, Hilden, Germany). The amount of collected RNA samples were measured by Cytation 5 plate reader and 800 ng (HBMEC) and 280 ng (HA) of RNA were reverse-transcribed into cDNA with T100 Thermal Cycler (Bio-Rad, Hercules, CA, USA) using High-capacity cDNA Reverse Transcription kit (Applied Biosystems, Foster City, CA, USA). To analyze endothelial specific gene expressions in HBMECs ($n = 3$), microfluidic qRT-PCR was performed with Flex Six IFC (Fluidigm, South San Francisco, CA, USA) using the Fluidigm Biomark system (Fluidigm). To analyze glial reactivity of HA in 2D and 3D culture system ($n = 4$), standard qRT-PCR was performed with a StepOnePlus Real-Time PCR system (Applied Biosystems) using TaqMan Fast Universal PCR Master Mix (Applied Biosystems). The target genes were assessed using commercially

available primers (GFAP: Hs00909233_m1, VIM: Hs00958111_m1, LCN2: Hs01008571_m1, VWF: Hs01109446_m1, SELE: Hs00174057_m1, PECAM1: Hs01065279_m1, VECAD (CDH5): Hs00901465_m1, OCLN: Hs00170162_m1, ZO-1 (TJP1): Hs01551861_m1, CAT1 (SLC7A1): Hs00931450_m1, LAT1 (SLC7A5): Hs00185826_m1, OCT1 (SLC22A1): Hs00427552_m1, GLUT1 (SLC2A1): Hs00892681_m1, CERP (ABCA1): Hs01059137_m1, P-GP (ABCB1): Hs00184500_m1, MRP1 (ABCC1): Hs01561483_m1, LRP1: Hs00233856_m1, AGER: Hs00542584_g1, ICAM1: Hs00164932_m1, VCAM1: Hs01003372_m1; Applied Biosystems; All primers are also listed in Supplementary Table 1). The results were quantified by the comparative $C_t$ method. $C_t$ values for samples were normalized to the expression of the housekeeping gene, glyceraldehyde 3-phosphate dehydrogenase (GAPDH: Hs02786624_g1; Applied Biosystems).

**Immunocytochemistry**. To visualize cell-specific marker expression, immunocytochemistry was performed. Briefly, samples were fixed with 2% paraformaldehyde (PFA; Santa Cruz Biotechnology, San Diego, CA, USA) for 15 min at RT. After permeabilizing in 0.1% Triton X (Sigma-Aldrich) in PBS for 15 min, the samples were blocked with 2% bovine serum albumin (BSA; Sigma-Aldrich) in PBS for 1 h at RT. Subsequently, the samples were incubated with primary antibodies at 4 °C overnight, washed three times with 1% BSA. The following antibodies were used for immunocytochemistry: goat anti-ZO-1 (1:200; Abcam, Cambirdge, MA, USA), mouse anti-GFAP (1:200; Invitrogen, Carlsbad, CA, USA), rabbit anti-S100β (1:200; Invitrogen), rabbit anti-Phospho eNOS (Ser1177) (1:200; Invitrogen), AlexaFluor 488 conjugated rabbit anti-α-SMA (1:200; Abcam), and rabbit anti-AQP4 (1:200; Invitrogen). Then samples were incubated with fluorescence-conjugated secondary antibodies (donkey anti-goat AlexaFluor 633 (1:200; Invitrogen), chicken anti-mouse AlexaFluor 594 (1:200; Invitrogen), and donkey anti-rabbit AlexaFluor 647 (1:200; Abcam)) for 6 h at 4 °C to visualize the targets. Nuclei were counterstained with 4,6-diamino-2-phenylindole (DAPI; Invitrogen) and stored in PBS before imaging. Fluorescently visualized samples were examined using a confocal microscope (LSM 700, Carl Zeiss, Oberkochen, Germany).

**Permeability measurement**. After 60 h of culture, culture medium containing 500 µg mL$^{-1}$ of 4 kDa or 40 kDa FITC-dextran (Sigma-Aldrich) was introduced into the luminal channel of the device at 16 µL min$^{-1}$ with a PhD Ultra syringe pump (Harvard Apparatus). Simultaneously, culture medium from one side channel was sampled at 4 µL min$^{-1}$ with a syringe pump for 1 h. Fluorescence intensities of 500 µg mL$^{-1}$ of FITC-dextran solution and the sampled solutions are measured using a Cytation 5 plate reader ($n = 4$ for each condition). The dextran concentrations in the solutions were calculated with the measured fluorescence intensity values using a standard calibration curve. Permeability coefficients were calculated using the following equation:

$$P = V \frac{\frac{dC}{dt}}{\Delta C} \tag{1}$$

Where $V$ is the volume of the sampled solution, $A$ is the surface area of the endothelial barrier, $\frac{dC}{dt}$ is the concentration change in the abluminal space along time, and $\Delta C$ is the concentration difference across the barrier.

**TEER measurement**. The TEER of the endothelial monolayer formed in the device was measured using a custom electrode adapter[30,65] made with Rj11 plug and Ag, Ag/AgCl electrode wires (381 µm in diameter and 3 cm in length, A-M Systems, Sequim, WA, USA) connected to EVOM2 volt-ohmmeter (Word Precision Instruments, Sarasota, FL, USA) which generates a constant 10 µA of AC current at 12.5 Hz while measuring resistance. To reduce background resistance and error, the electrode wires were placed in a tygon tubing (1/32"ID x 3/32"OD, Cole-Parmer, Vernon Hills, IL, USA) filled with culture medium and inserted into the channels (Supplementary Fig. 11). After 1 min of stabilization, 5 multiple readings were averaged for each chip. To calculate TEER, the measurements from the chips in the absence of the cells were subtracted from the resistance of each device, and then the values were multiplied by the surface area of endothelial monolayer overlapping with the lower channel (0.015 cm$^2$).

**Image analysis**. Quantitative analysis of cell distribution, cell morphology, and AQP4 polarization were performed using ImageJ (NIH, Bethesda, MD, USA). For cell distribution analysis, fluorescence intensity profile of each color (red, green, and white) was analyzed using Matlab (Mathworks, Natick, MA, USA). To analyze the morphology of HAs, the boundaries of cells were obtained automatically using magic wand tool on the maximal intensity projection image. Distribution of AQP4 was quantified by measuring the fluorescence intensity profile along the z-axis in z-stack images of the perivascular channel using ImageJ. The lower channel was divided into the two spaces (top half—vascular side, bottom half—parenchymal side) and the average of the fluorescence intensity from each space was calculated. The averaged intensity within the top half space (vascular side) was divided by the averaged intensity within the bottom half space (parenchymal side) to calculate the AQP4 polarization index.

**Nanoparticle synthesis**. The microvortex propagation mixer (µVPM) that we reported previously[63] was used for the synthesis. Briefly, the µVPM was connected

to syringe pumps (Harvard Apparatus) to introduce the solutions into the device. The precursor solutions including a lipid solution that was composed of 1,2-dimyristoyl-sn-glycero-3-phosphocholine (DMPC; Avanti Polar Lipids, Alabaster, AL, USA) with a concentration of 2.75 mg mL$^{-1}$ in ethanol, and a apolipoprotein A1 from human plasma (Sigma-Aldrich) with a concentration of 0.2 mg mL$^{-1}$ in PBS were added into the mixer. The flow ratio between the side streams and the center stream in the mixer was 5.5:1. The mixed solution was collected and then washed three times with PBS using a 10,000 M.W. centrifugal filter (EMD Milli-pore, Darmstadt, Germany) at a speed of 2585 × g for 20 min. The size distribution of the final sample was measured with a Zetasizer Nano ZS (Malvern Instruments, Worcestershire, United Kingdom). Fluorescently labeled eHNPs were synthesized with 1,1′-Dioctadecyl-3,3,3′,3′-Tetramethylindotricarbocyanine Iodide (DiR′; Invitrogen) or a modified lipid precursor solution containing 15% (w/w) 1,2-dimyristoyl-sn-glycero-3-phosphoethanolamine-N-(lissamine rhodamine B sulfo-nyl) (Rhodamine-DMPE; Sigma-Aldrich). The amount of protein of the final sample was quantified using the Micro BCA Protein Assay kit (Invitrogen).

**Biodistribution study**. All animal experiments were reviewed and approved by the Georgia Tech's Institutional Animal Care and Use Committee. 4–5 week-old male balb/c mice (Jackson Labs, city, state, USA) were given an irradiated dietary regiment until the mice were 21–22 weeks of age. For biodistribution study, 1 mg kg$^{-1}$ of eHNP-A1 was systemically administered to the mouse via tail vein injection. Injection of 200 µL saline was served as control. 24 h after administration, mice were sacrificed and perfused with saline and 4% PFA for 15 min. Then organs (brain, heart, lung, liver, kidneys, and spleen) were harvested to visualize their DiR content using an in vivo imaging system (IVIS; Perkin Elmer, Waltham, MA, USA). To visualize the eHNP-A1 internalization inside the brain tissue, the har-vested brain tissues were cryosectioned into 10 µm slices and stained with DAPI using the DAPI-containing antifade mounting medium (H-1200; Vector Labora-tories, Burlingame, CA, USA). The slides were then imaged under a confocal microscope (Zeiss LSM 780).

**Nanoparticle distribution study in a chip**. Culture medium containing eHNP-A1 at a concentration of 10 µg mL$^{-1}$ was introduced into the upper channel of the device. To block the NP transport via SR-B1, 200 µM of BLT-1 (Sigma-Aldrich) was treated in the upper channel for 1 h prior to the NP incubation. Following 2 h of NP incubation with culture medium with or without BLT-1 (200 µM), 10 µL of culture medium was sampled from the upper channel. The upper channel was washed with PBS and filled with Dispase (Corning) to digest Matrigel in the lower channel. After 30 min, 10 µL of the mixture of the culture medium and digested Matrigel from the lower channels were collected. To measure the concentrations of nanoparticles, fluorescence intensities of the culture medium injected into the upper channel, sampled from the upper channel ($n = 14$ for control and $n = 13$ for BLT-1), and the culture medium sampled from the lower channels ($n = 7$ for control and $n = 5$ for BLT-1) were measured using the Cytation 5 plate reader.

**Nanoparticle uptake study using flow cytometric analysis**. Prior to seeding cells into the device, HBMECs and HAs were fluorescently labeled with CellTracker (Deep Red and Green CMFDA, respectively; Invitrogen) as per manufacturer's instructions. After 2 h of 10 µg mL$^{-1}$ of NP incubation in the upper channel of the device, the devices were washed three times with PBS and 0.25% Trypsin-EDTA (Invitrogen) and Dispase (Corning) were injected into the upper channel and lower side channels, respectively. HBMECs and HAs were then collected in culture medium and spun down at 200 × g for 5 min. Then cell mixture was fixed in 4% PFA (Santa Cruz Biotechnology) for 30 min and stored in ice-cold FACS buffer (PBS with 2% FBS (Invitrogen)) at 4 °C. The HBMEC and HA fractions in total sample ($n = 5$ for control and $n = 6$ for BLT-1) were obtained by BD FACSAria III Cell Sorter (BD Biosciences, San Jose, CA, USA) and quantitatively analyzed using FlowJo software (FlowJo LLC, Ashland, OR, USA). Debris and doublets were excluded using FSC-A/SSC-A, FSC-A/FSC-W, respectively. The cells were then gated for the appropriate markers for the cell type (Supplementary Fig. 12).

**Calculation of nanoparticle distribution on a chip**. Loss of the injected NPs in a device due to their non-specific adsorption to the PDMS surface was measured by sampling the NP solution after 2 h of incubation in a device consisting of a single-layer of the vascular channel. The NP distribution in HBMECs was calculated as follows:

$$(\textbf{HBMEC uptake}_{\textbf{calculation}}) = (\textbf{Total injected}\,(\textbf{100\%})_{\textbf{Sampling}}) - (\textbf{Loss}_{\textbf{sampling}})$$
$$- (\textbf{Vascular}_{\textbf{Sampling}}) - (\textbf{Perivascular}_{\textbf{Sampling}})$$

(2)

Where (Vascular$_{Sampling}$), (Perivascular$_{Sampling}$), and (Total injeced)(100%)$_{Sampling}$) are fluorescence intensities of NP solutions sampled from the vascular channel, perivascular channel, and injected solution, (Loss$_{sampling}$) is the loss of NPs in a device measured by sampling, and (HBMEC uptake$_{calulation}$) is percentages calculated by the equation.

**Statistics analyses**. Data are presented as mean ± standard error of the mean (s.e.m.) unless otherwise stated. All statistical analyses were done using GraphPad Prism 6 (GraphPad Software, La Jolla, CA). Unpaired Student's $t$-tests was applied and $p$-values of < 0.05, 0.01, 0.005, and 0.001 were considered significant in all tests. All chip experiments were reproduced for at least two times to confirm data reliability. Per experiment, at least two biological and technical replicates were used.

**Reporting Summary**. Further information on research design is available in the Nature Research Reporting Summary linked to this article.

## Data availability
All data generated or analyzed during this study are included in this published article (and its supplementary information files), or are available from the corresponding author on reasonable request.

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

## Acknowledgements

This work was supported by the National Institutes of Health Director's New Innovator Award 1DP2HL142050 (Y.K.), National Institute of Neurological Disorders and Stroke (NINDS) R21NS091682 (Y.K.), and the National Institutes on Aging R21AG056781 (Y.K.). The content is solely the responsibility of the authors and does not necessarily represent the official views of the National Institutes of Health. We thank the core facilities at the Parker H. Petit Institute for Bioengineering and Bioscience, and the Institute for Electronics and Nanotechnology at Georgia Institute of Technology, a member of the National Nanotechnology Coordinated Infrastructure, which is supported by the National Science Foundation (ECCS1542174).

## Author contributions

S.I.A. and Y.K. developed the device, designed experiments, and wrote the manuscript. S.I.A performed experiments and analyzed data. Y.J.S performed in vivo animal studies. H-J.P. performed RT-qPCR experiments. J.K. synthesized and characterized nano-particles. J.J.C. and S.I.A. performed flow cytometry. S.I.A. and Y.R. fabricated micro-fluidic devices. A.I.L., T.J.M. and H-J.S. provided insightful comments in the overall project. Y.K. conceived, initiated, and supervised the overall project.

## Competing interests

The authors declare the following competing financial interests: In compliance with the institutional guidelines of the Georgia Institute of Technology, Y.K. discloses his financial interest in Mepsgen and Mepsgenlab, two biotechnology companies developing micro-engineered physiological systems and biomimetic nanoparticles for medical applications. Mepsgen and Mepsgenlab did not support the aforementioned research, and currently these companies have no rights to any technology or intellectual property developed as part of this research. Y.K. declares no non-financial competing interests. All the other authors declare no competing interests.
