## [Peer Review File · Nature Communications]

Reviewers' comments:

Reviewer #1 (Remarks to the Author):

The authors show a microengineered BBB model with three different human cell types in it and were able to measure some functional properties of blood-brain barrier. Although this is very nice and the biological work seems of high quality, part of the more technical work needs more clarification. Therefore I do not think it can be published in the present form and the manuscript needs some adaptations before (re)submission.

Comments:

- In the introduction the authors refer to fig1a and mentioning the cells present there. However endothelial cells are not mentioned in the figure.
- At the end of the introduction, the authors mention that they use hBMECs so that it is more biological relevant. I agree with this. However they compare their model with other recent tri-culture models, without stating what the cells are used in these ones. Therefore it is difficult to assess how innovative or new the model of the authors is. Besides that, why do the authors not use hCMEC/d3?
- In the first paragraph of the result section, 'fluid flow', 'porous membrane' and '3D hydrogel' are mentioned without clarifying these. So please indicate what the fluid flow and shear stress is, give the pore density and pore size of the membrane and also the thickness, what kind of hydrogel is used including its concentration.
- They also show in fig1h a model. For me it is not clear how they take the cell layer and membrane into account. Can the authors clarify this?
- 'high-throughput' is mentioned, but quantify this and also relate this to other high-throughput systems mentioned in the literature (put this in the discussion)
- The authors refer to fig 1l (later it is also figure 3 mentioned) and clearly state that you can see AQP4 expression underneath the porous membrane. I cannot see this. How is the quantification done?
- Page 7 membrane -> membrane
- Caption fig1 MTS is mentioned. Please clarify what this is.
- How is TEER measured? The authors claim that on page 9, but it is not clear how this is done. In the methods section they state that they use Ag/AgCl electrodes; However the measured value is also determined by the position of the electrodes and the length of the microchannel (that can also have an enormous resistance. Furthermore the potentials across the membrane should be homogeneous, otherwise the influence of the TEER on the position of the cells is different leading to a higher influence of the first part on the measured value. Also what will a pore do? Additionally the frequency, voltage and data (subtraction of controls etc) should be more clearly described.
- They authors say that blocking of SR-B1 lead to more NPs in the vascular channel and this is a nice result. However you would also expect that less NPs end up in the perivascular channel and this is not the case. How is this possible? Are there pores in the cell layer?
- Fig4. 'in vivo' needs to be cursive. Please also indicate 1 2 3 in figure 4m and for me figure 4q is not clear. What do we see?
- In the discussion 'high reproducibility' is mentioned. Please quantify?
- Please also discuss the results from previous literature on hCMEC/d3. Now they state that most use HUVECS but there are also other cell types used.

Reviewer #2 (Remarks to the Author):

The manuscript contains data on establishment of physiologically relevant in vitro BBB model suitable for pharmacokinetic studies. It is well-designed and well-written study, however, in the current form it might not affect significantly further progress in the corresponding area of research. There are several examples of almost similar approaches to development of in vitro models of the neurovascular unit, i.e. described by B.T. Hawkins et al., 2015; A. Herland et al., 2016; M.A. Kaisar et al., 2017 etc. So, better confirmation of advantages of this original BBB model over existing ones is highly recommended.

There are some points that should be corrected or detailed by the authors:

1. The ratio of different cell types (endothelial and non-endothelial) could greatly affect the integrity and functional properties of the BBB in in vitro models. So, the optimal ratio of cells that was achieved in experiments should be indicated.
2. What pericytes markers were used for pericytes phenotyping?
3. Regarding astroglial markers, why GFAP only was selected as a reliable marker? Actually, it is rather non-specific in the context of perivascular astroglia, and, for example, s100beta might be more accurate for phenotyping perivascular astrocytes.
4. Morphology of perivascular astroglia within the neurovascular unit depends on functional status of astrocytes, i.e. it might be changed due to activation of mechanisms of astrocyte-controlled vessel tone. Thus, preserving "resting" morphology of astrocytes in 3D Matrigel throughout their co-culturing with endothelial cells and pericytes might not be a positive sign for reconstruction of functionally competent BBB.
5. Authors claimed that water exchange through the brain microvessel endothelial cells monolayer co-cultured with pericytes and astrocytes was preserved due to correct expression of AQP4. Do they have any experimental data confirming functional activity of AQP4 in the in vitro BBB model?
6. Resistance of endothelial cells to shear stress was tested in physiological conditions (4 dynes/cm²), however, rather complicated (multichannel) construction of the microfluidic chamber might require more experimental data on functional competence of endothelial layer (i.e. testing effects of 1 and 10 dynes/cm²) as well as data on water exchange (as indicated in point 5).
7. Some terms in the text should be corrected, i.e. LRP1 is not a lipoprotein receptor protein-1, but LDL lipoprotein receptor-related protein-1).
8. Why microglia medium was used in a series of experiments (supp. fig. 1)?
9. The obtained value of TEER (200 for BBB model) seems to be rather low comparing with other microfluidic BBB models (i.e. Y.I. Wang et al., 2017). Also, it is not clear whether shear stress was able to increase TEER.
10. Fig. 1i contains data obtained with MTS assay. Authors should specify in the text and in the figure legend what parameter they actually measured: viability or metabolic activity.

Responses to Reviewers

Reviewer #1

Comment 1. In the introduction the authors refer to fig1a and mentioning the cells present there. However endothelial cells are not mentioned in the figure.

Response: Thank you for pointing out that endothelial cells are not mentioned in the figure. We added “endothelial cells (ECs)” in Figure 1a.

Added text:

On page 7 – Figure 1a - EC

On page 8, line 137 (Figure caption for Fig.1a) - Endothelial cells (ECs)

Comment 2. At the end of the introduction, the authors mention that they use hBMECs so that it is more biological relevant. I agree with this. However, they compare their model with other recent tri-culture models, without stating what the cells are used in these ones. Therefore, it is difficult to assess how innovative or new the model of the authors is. Besides that, why do the authors not use hCMEC/d3?

Response: Previous studies have indicated that HBMEC is the most suitable and promising cell line for a human *in vitro* BBB model in terms of the barrier tightness and permeability among the four available immortalized human brain capillary endothelial cell lines (hCMEC/D3, HBMEC, TY10, and BB19) (Eigenmann et al., 2013). Additionally, in our present manuscript, we have confirmed and validated the BBB specific gene expressions of HBMEC when co-cultured with primary human brain vascular pericytes (HBVP) and human primary astrocytes (HA), demonstrating the capability of HBMEC to construct a BBB model. For your information, we added below a table showing the brain specificity and human origin of endothelial cell types used in other recent BBB tri-culture models. In addition, as suggested we have stated the endothelial cell types used in the previous studies in the revised manuscript (Discussion).

Cell type	Brain specificity	Human specificity	Refs	Validation of specialization
b.EnD3	O	X	(Wang et al., 2016)	X
			(Booth and Kim, 2012)	X
HUVEC	X	O	(Bang et al., 2017)	X
			(Adriani et al., 2017)	X
hCMEC/D3	O	O	(Adriani et al., 2017)	X
Primary hBMEC	O	O	(Herland et al., 2016)	X
			(Maoz et al., 2018)	X

Added text:

On page 21, line 359 – Immortalized human brain endothelial cells are good candidates for standardized screenings due to their availability as well as indefinite proliferation while preserving their properties. Previous studies using hCMEC/D3 in microfluidic BBB models showed their barrier function with junctional protein expression, TEER, and permeability. In our present study, we have used HBMEC as it has been reported that

HBMEC is the most suitable and promising immortalized human brain endothelial cell line among the four available cell lines (hCMEC/D3, HBMEC, TY10, and BB19) for *in vitro* modeling of human BBB in terms of the barrier tightness and permeability.

Comment 3. In the first paragraph of the result section, 'fluid flow', 'porous membrane' and '3D hydrogel' are mentioned without clarifying these. So please indicate what the fluid flow and shear stress is, give the pore density and pore size of the membrane and also the thickness, what kind of hydrogel is used including its concentration.

Response: We have added the detailed information of the fluid flow (16 $\mu\text{L}/\text{min}$ for shear stress (4 dyne/cm^2)), the membrane properties (pore density (1E5 pores/ cm^2), pore size (8 μm), and thickness (7 μm)), and the hydrogel (growth factor reduced Matrigel with concentration (5 mg/mL)) in the revised manuscript (Results).

Added text:

On page 6, line 105 -The upper layer of the device mimics the vascular space of the brain microvasculature where an endothelial monolayer is formed on a 7 μm thick porous membrane (8 μm diameter pores at a density of 1E5 pores/ cm^2) with 16 $\mu\text{L}/\text{min}$ of continuous fluid flow (shear stress: 4 dyne/cm^2). The lower layer accommodates pericytes underneath the membrane and astrocytes in a 3D Matrigel (5 mg/mL) in the center channel along with the two side channels (Fig. 1f).

Comment 4. They also show in fig1h a model. For me it is not clear how they take the cell layer and membrane into account. Can the authors clarify this?

Response: We have conducted computational fluid dynamics simulation as shown in Figure 1h to better design such a complex microfluidic BBB model to have sufficient diffusive transport of culture medium components within a period of medium refresh time. In our computational model, we defined the porous membrane as a porous medium with the properties provided by the manufacturer (e.g. thickness, pore size, and pore density) but we did not consider accurate transport mechanisms by biological factors like the endothelial cell layer in our model. In addition, we have pointed out the rationale why we performed the simulations in the revised figures (Fig. 1h and Supplementary Fig. 1). We added the detailed information for our computational model in the revised manuscript (Methods).

Added text:

On page 6, line 115 - The device is designed to have sufficient diffusive transport of culture medium components into the hydrogel channel within 1 hour of media refreshment in the upper and the two side channels (Fig. 1h and Supplementary Fig. 1).

On page 25, line 441 - Matrigel and the porous membrane were defined as a porous medium with porosity value of 0.8 and 0.016, respectively. The computational simulation was performed within a microfluidic device without cellular components.

Comment 5. 'high-throughput' is mentioned, but quantify this and also relate this to other high-throughput systems mentioned in the literature (put this in the discussion)

Response: One major advantage of microfluidics-based organ-on-a-chip models is the ability to leverage well-established microfluidic technologies that allow for the high-throughput configuration (Hung et al., 2005; Thorsen et al., 2002; White et al., 2011). In our present study, the BBB model allowed for multiple analyses including TEER measurement, nanoparticle sampling, and FACS analysis while the multiple devices were regulated by multi-syringe racks at the same time. Moreover, our group has the ability to parallelize microfluidic reactors for building a high-throughput system as demonstrated in our recent approach that showcased a microfluidic parallelization technology that integrates 25 reactors while preserving advantages of the microscale engineering (Toth et al., 2017), which can be combined to increase throughput of the BBB model. We have included our discussion of the ‘high-throughput’ in the revised manuscript (Discussion).

Added text:

On page 24, line 411 – Our BBB model allowed for multiple analyses including TEER measurement, nanoparticle sampling, and FACS analysis while the multiple devices were regulated by multi-syringe racks at the same time (**Supplementary Fig. 4a**). Microfluidic parallelization technology that integrates multiple devices while preserving advantages of the microscale organ-on-a-chip engineering will further provide higher throughput system (Toth et al., 2017).

Comment 6. The authors refer to fig 1l (later it is also figure 3 mentioned) and clearly state that you can see AQP4 expression underneath the porous membrane. I cannot see this. How is the quantification done?

Response: First, we admit that creating a section view from 3D confocal z-stack images may not often provide very clear images, although we did our best by using confocal microscopy to quantify the relative expressions of AQP4 across the vertical axis of the lower channel as shown in Figure 3j. We quantified AQP4 polarization by dividing the AQP4 expression of the vascular side towards endothelium by that of the parenchymal side. This quantification method was also used by other groups (Eidsvaag et al., 2017). Briefly, we first measured the fluorescence intensity profile along the z-axis in z-stack images of the perivascular channel using ImageJ. The lower channel was divided into the two spaces (top half – vascular side, bottom half – parenchymal side) and the average of the fluorescence intensity from each space was calculated. The averaged intensity within the top half space (vascular side) was divided by the averaged normalized intensity within the bottom half space (parenchymal side) to calculate the AQP4 polarization index.

In addition, with the reviewer’s comment, we have made efforts to better show the AQP4 expression at the end-feet of astrocytes which are located underneath the porous membrane in our BBB model in Figure 1l as well as Figure 3i and 3j. First, we have reduced the relative intensity of GFAP (white) in the merged image of Figure 1l to better exhibit the AQP4 (yellow) expression underneath the membrane. We pointed out where the membrane is located in Figure 1l and 3h-j and what the red line represents in Figure 3h. Please note that AQP4 expressions in astrocytes are “polarized” when co-cultured with endothelial cells and pericytes in our BBB models compared to those when cultured alone, meaning that AQP4 expressions are also shown underneath the membrane, not underneath it only. Our BBB model mimics the polarized expression of AQP4 observed *in vivo*, whereas previous *in vitro* culture models have only shown spread distribution of AQP4 in plasma membrane of astrocytes (Di Benedetto et al.,

2016; Nicchia et al., 2000; Nicchia et al., 2003; Saadoun et al., 2005). We have also shown that we can reproduce these 2D spread distributions of AQP4 like these studies (see the image of 2D monoculture of astrocytes with their common AQP4 expression pattern (Supplementary Figure 8)).

More importantly, to further convince our astrocytic end-feet polarization in addition to AQP4 expressions, we have conducted additional experiments to observe another marker for astrocytic end-feet, α -syntrophin, which is a dystrophin associated protein that serves as the immediate anchor for AQP4 in end-feet membranes (Neely et al., 2001). We have inserted these new images to the revised Figures 1I and 3i and also added the detailed quantification methods in the revised manuscript (Methods).

Added text and images:

On page 7, Figure 1I – I, Aquaporin-4 (AQP4, yellow) and α -syntrophin (α -syn, magenta) expressions at astrocytic end-feet (GFAP, white) underneath a porous membrane (indicated as the dotted line) in the lower channel (Blue arrows indicate co-localization of AQP4 with α -syn.) (scale bar = 50 μ m).

On page 15, Figure 3h and 3i – h, Quantitative analysis of AQP4 polarization by measuring AQP4 distribution in vascular and parenchymal side in the perivascular channel. i, Co-localization of AQP4 (AQP4, yellow) and α -syntrophin (α -syn, magenta) at astrocytic end-feet (scale bars = 20 50 μ m).

On page 30, line 549 - Distribution of AQP4 was quantified by measuring the fluorescence intensity profile along the z-axis in z-stack images of the perivascular channel using ImageJ. The lower channel was divided into the two spaces (top half – vascular side, bottom half – parenchymal side) and the average of the fluorescence intensity from each space was calculated. The averaged intensity within the top half space (vascular side) was divided by the averaged normalized intensity within the bottom half space (parenchymal side) to calculate the AQP4 polarization index.

(Supplementary Information) On page 9 - **Supplementary Figure 8. Aquaporin-4 (AQP4) and α -syntrophin (α -syn) expression in 2D cultured astrocytes.** Confocal images of 2D monoculture of astrocytes (GFAP, white) showing diffusive expression of AQP4 (AQP4, yellow) and α -syn (α -syn, magenta), unlike their polarized expressions shown in the revised Figures 3h and 3i as above. Scale bars = 50 μ m.

Comment 7. Page 7 membrane -> membrane

Response: Thank you for pointing this out. We corrected the typo in the manuscript.

Comment 8. Caption fig1 MTS is mentioned. Please clarify what this is.

Response: MTS assay can be described as a one-step MTT assay, which is widely used to measure cell metabolic activity. In detail, MTS, (3-(4,5-dimethylthiazol-2-yl)-5-(3-carboxymethoxyphenyl)-2-(4-sulfophenyl)-2H-tetrazolium), is reduced by NAD(P)H-dependent oxidoreductases and dehydrogenases of metabolically active cells and produces aqueous, formazan product. The formazan product can be detected with its absorbance at 490 nm. Description and clarification of the MTS assay were added in the revised manuscript (Results and Methods).

Added text:

On page 6, line 122 - metabolic activity assays (see Methods for details)

On page 8, line 150 - (3-(4,5-dimethylthiazol-2-yl)-5-(3-carboxymethoxyphenyl)-2-(4-sulfophenyl)-2H-tetrazolium) (MTS) assay

On page 26, line 454 - To optimize cell culture condition, cell metabolic activity in different medium condition were quantified using a (3-(4,5-dimethylthiazol-2-yl)-5-(3-carboxymethoxyphenyl)-2-(4-sulfophenyl)-2H-tetrazolium) (MTS) assay (n = 6), which measures the formazan product by cell metabolic activity. CellTiter 96 AQueous One Solution Cell Proliferation Assay (Promega, Madison, WI, USA) was added to HBMECs, HBVPs, and HAs cultured in Endothelial cell medium (E), Astrocyte medium (A), Pericyte medium (P), Microglia medium (M; Sciencell), E+G (E:A:M= 1:1:1:1), and E+P+G (E:P:A:M=1:1:1:1) after 3 days of culture. Here, we describe 1:1 mixture of A and E as glial cell medium (G). After 4 h of incubation, the optical absorbance of each sample was measured at 490 nm using Cytation 5 plate reader (BioTek, Winooski, VT, USA).

Comment 9. How is TEER measured? The authors claim that on page 9, but it is not clear how this is done. In the methods section they state that they use Ag/AgCl electrodes; However, the measured value is also determined by the position of the electrodes and the length of the microchannel (that can also have an enormous resistance. Furthermore, the potentials across the membrane should be homogeneous, otherwise the influence of the TEER on the position of the cells is different leading to a higher influence of the first part on the measured value. Also what will a pore do? Additionally, the frequency, voltage and data (subtraction of controls etc) should be more clearly described.

Response: TEER was measured following a protocol as previously reported (Kim et al., 2014). We tried to reduce the possible background resistance and error by maintaining large area of the electrode wires exposed to the culture media through the device. Briefly, the electrode wires (381 μm in diameter and 3 cm in length) were placed in a tygon tubing filled with culture medium and inserted into the channels. One end of the electrode wires was positioned at the inlets and outlets of upper and lower side (right) channels and the other end was connected to a custom connector to fit EVOM2 volt-ohmmeter (World Precision Instruments, Sarasota, FL, USA). TEER measurements were taken in a 4-point measurement method using the EVOM2 volt-ohmmeter that is commercially available and widely used to measure TEER as validated by several research articles (Booth and Kim, 2012; Jamieson et al., 2019; Park et al., 2019). The EVOM2 generates a constant 10 μA of AC current at 12.5 Hz while measuring the resistance. Moreover, we made an effort to minimize the error by averaging the TEER values from 5 multiple readings for each device. All TEER values ($\text{ohm}\cdot\text{cm}^2$) were based on the surface area of the endothelial monolayer in our BBB model, an area that overlaps that of the lower channel (0.015 cm^2). Regarding the role of the pore of the membrane, we have selected the most permeable membrane (1000 mL/min/ cm^2 of water flow rate across the membrane at 10 psi) to minimize the effect of the pore on the potential background resistance increase, as well as better perfused model construction.

In this revision, we have also done more experiments to add more data points to convince the reproducibility of our measurement and also clearly described how to measure TEER in detail in the revised manuscript (Figure 2I and Methods).

Added text:

On page 11, Figure 2I - I, TEER measured across the membrane between the upper and lower layers without cells (No cell – hydrogel only), with an endothelial monolayer (EC), and an endothelial monolayer with pericytes and astrocytes (BBB) (n=6 for No cell, n=12 for EC and BBB, **p<0.01 and ****p<0.001).

On page 30, line 534 - The TEER of the endothelial monolayer formed in the device was measured using Ag/AgCl electrode wires (381 μm in diameter and 3 cm in length, A-M Systems, Sequim, WA, USA) connected to EVOM2 volt-ohmmeter (Word Precision Instruments, Sarasota, FL, USA) which generates a constant 10 μA of AC current at 12.5 Hz while measuring resistance. To reduce background resistance and error, the electrode wires were placed in a tygon tubing (1/32"ID x 3/32"OD, Cole-Parmer, Vernon Hills, IL, USA) filled with culture medium and inserted into the channels. After 1 min of stabilization, 5 multiple readings were averaged for each device. To calculate TEER, the measured resistance was multiplied by the surface area of the endothelial monolayer overlapping with the lower channel (0.015 cm^2).

Comment 10. They authors say that blocking of SR-B1 lead to more NPs in the vascular channel and this is a nice result. However, you would also expect that less NPs end up in the perivascular channel and this is not the case. How is this possible? Are there pores in the cell layer?

Response: Brain endothelial cells are known to express several receptors that interact with HDL, including SR-B1 (Fung et al., 2017), LDLR (Lillis et al., 2008), and LRP1 (Storck et al., 2016), indicating that there are still other transport mechanisms with which HDL can cross the BBB, although the amount might not be large like that by SR-B1. Moreover, blocking SR-B1 activity clearly reduced HDL uptake, whereas it may induce upregulation of other receptors mediating transcytosis or other determinants of endothelial transcytosis such as SNAREs, resulting in compensated transcytosis to a certain degree (Fung et al., 2017; Van Eck et al., 2003). Moreover, we confirmed that there was no other pore or gap in our BBB endothelial layer, since the BBB has a tight barrier function with the permeability coefficient for 4 kDa FITC-dextran ($6.5\text{E-}7$ cm/s) in a similar level of that measured *in vivo* (Yuan et al., 2009). The molecular size of 4 kDa FITC-dextran is smaller than the eHNP-A1, and thus, we expect that the paracellular transport of eHNP-A1 was negligible. We explained these points in the revised manuscript (Result).

Added text:

On page 17, line 306 –This result indicates that blocking SR-B1 activity reduces eHNP-A1 uptake by HBMECs, whereas it may induce a compensatory upregulation of other receptors or determinants of endothelial transcytosis such as SNAREs (Fung et al., 2017; Van Eck et al., 2003).

Comment 11. Fig4. ‘in vivo’ needs to be cursive. Please also indicate 1 2 3 in figure 4m and for me figure 4q is not clear. What do we see?

Response: We have revised as commented by the reviewer. Figure 4q shows the quantitative numbers of cellular uptake of eHNP-A1 measured by FACS analysis as shown in figure 4o and p. Three devices for each group were analyzed with FACS to quantify the cellular uptake. We added the number of devices as well as the details in the revised figure caption.

Added text:

On page 20, line 339 (Figure caption for Fig. 4q) - Cellular uptake of eHNP-A1 in the BBB chip quantified from FACS analysis (n=3).

Comment 12. In the discussion 'high reproducibility' is mentioned. Please quantify?

Response: The reproducibility of *in vitro* experiments is related to regulation of the cell condition and culture environment. In our entire experiments, we have used HBMECs with passages in a range between 5~10, primary HAs and HBVPs with passages of 3~5 to maintain the high reproducibility. Moreover, our microfluidic system to control the microenvironment with high precision (Kim et al., 2011; Toth et al., 2018) compared to the conventional macroscale models has also contributed to the high reproducibility of mimicking the BBB structure and function. The high reproducibility of our system can be also shown in the low variability of our experimental results (e.g., TEER, permeability, and NP sampling).

Comment 13. Please also discuss the results from previous literature on hCMEC/d3. Now they state that most use HUVECS but there are also other cell types used.

Response: As described in our response to **Comment 2** above, we have added the results from previous literature which used other endothelial cell types in the revised manuscript (Discussion).

Reviewer #2

General Comments:

The manuscript contains data on establishment of physiologically relevant *in vitro* BBB model suitable for pharmacokinetic studies. It is well-designed and well-written study, however, in the current form it might not affect significantly further progress in the corresponding area of research. There are several examples of almost similar approaches to development of *in vitro* models of the neurovascular unit, i.e. described by B.T. Hawkins et al., 2015; A. Herland et al., 2016; M.A. Kaisar et al., 2017 etc. So, better confirmation of advantages of this original BBB model over existing ones is highly recommended.

Response: Our microengineered human blood-brain barrier (BBB) model is the first *in vitro* human BBB model that mimics the physiological characteristics of human nonreactive (resting) or reactive (activated) astrocytes in 3D and that for the first time enables quantitative analysis of nanoparticle transport across the BBB at tissue and cell levels. This modeling feature of controlled 3D astrocytes in a BBB model is extremely important with the current in-depth understanding of the critical contributions of astrocytes to neurodegenerative diseases, as well as BBB construction and disruption in CNS diseases. Importantly, recent studies in high impact journals (DeStefano et al., 2018; Park et al., 2018; Placone et al., 2015; Yi et al., 2015) have reported that not only the barrier function of the brain endothelium but also the phenotypic characteristics of perivascular cells like astrocytes and microglia have to be well recapitulated to develop an *in vitro* human BBB model. Our current model incorporates star-shaped astrocytes with their end-feet extending to the vascular base membrane in a quiescent condition, as well as establishing a tight endothelial barrier.

With the reviewer's comment, here we point out the advantages of our BBB model over existing ones (also see a table below). (Booth and Kim, 2012) and (Cho et al., 2015) showed *in vitro* BBB models with non-human species such as rat or mouse, limiting the models in addressing species difference between humans and other animals. (Herland et al., 2016) failed to demonstrate a physiologically relevant tri-culture model. (Bang et al., 2017) and (Campisi et al., 2018) showed a 3D tri-culture of brain endothelial cells, pericytes, and astrocytes; however, it remains difficult to validate the barrier function using TEER and assess the vascular and perivascular spaces separately to quantify the molecular distributions. (Park et al., 2019) and (Vatine et al., 2019) showed double-layered culture systems like transwell models with technical advantages to allow real-time measurement of TEER and direct access to vascular and perivascular space for quantitative assessment of barrier function; however, these models remained restricted to 2D astrocyte culture.

Refs	Triculture (E+A+P)	Human cell source	3D culture of astrocytes	Shear flow
Booth and Kim, Lab Chip, 2012	x	x	x	o
Cho et al., Sci.Rep., 2015	x	x	o	x
Hawkins et al., Brain.Res., 2015	x	x	o	x
Herland et al., PLoS One, 2016	x	o	o	x
Bang et al., Sci.Rep., 2017	x	x	o	x
Campisi et al., Biomat., 2018	o	o	o	x
Vatine et al., Cell stem cell, 2019	o	o	x	o
Park et al., Nat.Comm., 2019	o	o	x	o
Our model	o	o	o	o

Specific Comments:

Comment 1. The ratio of different cell types (endothelial and non-endothelial) could greatly affect the integrity and functional properties of the BBB in in vitro models. So, the optimal ratio of cells that was achieved in experiments should be indicated.

Response: We agree with the reviewer that the cell number ratio of difference cell types is critical in the development of in vitro models. The ratio between endothelial cells and pericytes used in our BBB model is 1.5:1, which was set based on our culture condition optimization, as well as on references stating that the ratio is estimated between 1:1 and 3:1 (Daneman and Prat, 2015). Although the ratio between astrocytes and other BBB cells is not clearly known yet, astrocytes have been observed to cover nearly the entire abluminal surface of healthy cerebral microvessels (Pardridge, 2002). That said, we have astrocytes 3D cultured in our BBB model in a culture density (1E7 cells/mL) that can cover most of the abluminal surface area (endothelial cells:astrocytes = 2:1). The final ratio of cells is added in the revised manuscript (Methods).

Added text:

On page 27, line 480 – The final cell number ratio between HBMECs and HBVPs in a device was 1.5:1, and the ratio between HBMECs and HAs was 2:1, which was optimized for HAs to cover ~99% of the perivascular surface of the endothelium.

Comment 2. What pericytes markers were used for pericytes phenotyping?

Response: We used alpha-smooth muscle actin (α -SMA) which has been widely used in previous studies (Campisi et al., 2018; Herland et al., 2016; Vatine et al., 2019). This information is in the figure and figure captions, as well as Methods.

Comment 3. Regarding astroglial markers, why GFAP only was selected as a reliable marker? Actually, it is rather non-specific in the context of perivascular astroglia, and, for example, S100 β might be more accurate for phenotyping perivascular astrocytes.

Response: We selected GFAP as it is a widely used marker for astrocytes. We agree that S100 β is a marker to identify the perivascular astrocytes. With the reviewer's suggestion, we have conducted additional experiments to label S100 β on the astrocytes cultured in our BBB model. As previously reported (Jones et al., 2017; Steiner et al., 2007), we note that S100 β labels the cytoplasm and nuclei of astrocytes, whereas GFAP labels the cytoplasm and processes. These additional images have been added in the revised manuscript (Figure 2f) and Supplementary Information.

Added text and images:

On page 11, Figure 2h – **g,h**, Astrocytes with star-shaped morphology labeled with GFAP (GFAP, white) (**g**) and S100 β (S100 β , magenta) (**h**).

Comment 4. Morphology of perivascular astroglia within the neurovascular unit depends on functional status of astrocytes, i.e. it might be changed due to activation of mechanisms of astrocyte-controlled vessel tone. Thus, preserving "resting" morphology of astrocytes in 3D Matrigel throughout their co-culturing with endothelial cells and pericytes might not be a positive sign for reconstruction of functionally competent BBB.

Response: We agree that morphology of perivascular astrocytes changes under the pathological conditions. Indeed, we successfully reconstructed a functionally competent BBB model with controllable status of astrocytes from resting to reactive conditions (Please see Figure 3e, 3f, and 3g). This is one innovative feature of our BBB model where we have not intended to "maintain" "resting" morphology of astrocytes in 3D Matrigel in our BBB model, but rather demonstrate the ability to model "activated or reactive" astrocytes (for example, using IL-1 β in Figure 3g) from "resting" ones (Figure 3e) in the same BBB chip. Our model can clearly compare the distinct functions much more effectively than other existing ones that may have astrocytes pre-"activated" or "reactive" in their model with 2D astrocyte culture and thus are much less functionally competent BBB. With the reviewer's helpful comment, we clarified this view on the revised manuscript (Discussion).

Added text:

On page 16, line 267 (Figure caption for Fig.3e) - e, Gene expression of LCN2 in HAs cultured on Matrigel-coated 2D surface (f) and within 3D Matrigel (g) with IL-1 β treatment demonstrating the ability to model reactive astrocytes more effectively in 3D (n=4 for each condition, **p<0.01, ***p<0.005, and ****p<0.001).

On page 22, line 386 - with the ability to control status of astrocytes from resting to reactive conditions

Comment 5. Authors claimed that water exchange through the brain microvessel endothelial cells monolayer co-cultured with pericytes and astrocytes was preserved due to correct expression of AQP4. Do they have any experimental data confirming functional activity of AQP4 in the in vitro BBB model?

Response: As introduced in the manuscript, astrocytes in the perivascular space play a role in regulating the water homeostasis in the brain via the water channel protein AQP4 at their end-feet processes. Localization of AQP4 in astrocytic end-feet processes is therefore important to validate the successful recapitulation of the BBB in homeostatic and physiological conditions, as it clearly demonstrate the interaction between astrocytes and the vascular region. In this regard, we wanted to introduce the importance of AQP4 expression in polarized astrocytes in the reconstruction of a human BBB, which was overlooked in existing BBB models. In our BBB model, we used AQP4

as a marker for astrocytic end-feet to successfully show their coverage on an endothelial basal surface. However, experimental strategies to detect AQP4 function *in vitro* relies on a single cell level analysis (osmotic swelling assays) and microscopy techniques, which would be very difficult to apply for the measurement of the functional activity of AQP4 in the multicellular network constructed in our 3D BBB model.

Instead, in order to convince our successful reconstruction of polarized astrocytes, we have done additional experiments to further demonstrate the presence of the end-feet of astrocytes near the endothelium. We have successfully visualized additional marker, α -syntrophin, which is connected to AQP4 and controls the AQP4 polarization (Amiry-Moghaddam et al., 2003; Camassa et al., 2015; Nagelhus and Ottersen, 2013; Neely et al., 2001). We have included these images and additional new data in the revised manuscript (Figure 1l and 3i).

Added text:

On page 7, Figure 1l – l, Aquaporin-4 (AQP4, yellow) and α -syntrophin (α -syn, magenta) expressions at astrocytic end-feet (GFAP, white) underneath a porous membrane (indicated as the dotted line) in the lower channel (Blue arrows indicate co-localization of AQP4 with α -syn.) (scale bar = 50 μ m)

On page 14, line 248 - Our BBB model recapitulates a complex 3D network of HAs with expression of AQP4 along their branches while 2D culture models show diffusively expressed AQP4 in plasma membrane of astrocytes (**Supplementary Figure 8**). We analyzed AQP polarization by calculating the ratio of AQP4 labelled along astrocytic end-feet in a vascular side versus that in a parenchymal side of the perivascular channel as previously reported (Eidsvaag et al., 2017) (**Fig. 3h**) and by demonstrating co-localization with α -syntrophin, the immediate anchor for AQP4 that controls AQP4 polarization to astrocytic end-feet, in the model (**Fig. 3i**).

On page 15, Figure 3h,i – h, Quantitative analysis of AQP4 polarization by measuring AQP4 distribution in vascular and parenchymal side in the perivascular channel. i, Co-localization of AQP4 (yellow) and α -syntrophin (magenta) at astrocytic end-feet.

Comment 6. Resistance of endothelial cells to shear stress was tested in physiological conditions (4 dynes/cm^2), however, rather complicated (multichannel) construction of the microfluidic chamber might require more experimental data on functional competence of endothelial layer.

Response: To address the reviewer' comments, we have additionally tested the effects of different levels of shear stress on the endothelium with their gene expressions, TEER, and endothelial nitric oxide synthase (eNOS) phosphorylation. Our additional experiments showed that a physiological level of shear stress (4 dynes/cm^2) induced higher expressions of efflux transporter proteins p-gp and MRP1, higher TEER, and also eNOS phosphorylation. These new results have been added in the revised manuscript (Result) and Supplementary Information as below.

Added text:

On page 10, line 184– A physiological level of shear stress was responsible for inducing endothelial function with the barrier tightness (**Fig. 2I**), efflux transporter protein expression (**Supplementary Fig. 5**), and endothelial nitric oxide synthase (eNOS) phosphorylation (**Supplementary Fig. 6a,b**).

On page 11, Figure 2I – I, TEER measured from BBB models under different levels of shear stress ($n=5$ for No shear, $n=4$ for 0.4 dyne/cm^2 , and $n= 12$ for 4 dyne/cm^2 , $*p<0.05$)

(Supplementary Information) On page 6 - **Supplementary Figure 5**. Gene expressions of HBMECs in monoculture under static condition (Transwell) and physiological level of shear stress (chip, 4 dyne/cm^2).

(Supplementary Information) On page 7- **Supplementary Figure 6. Endothelial nitric oxide synthase (eNOS) phosphorylation in HBMECs under different levels of shear stress.** **a**, Confocal images of HBMECs from different shear stress conditions labelled with phospho-eNOS (Ser1177) (eNOS, green; DAPI, blue) (scale bars = 20 μm). **b**, Fluorescence intensities of eNOS from confocal images normalized by the average intensity of the static condition ($n=9$ for each condition, **** $p<0.001$).

Comment 7. Some terms in the text should be corrected, i.e. LRP1 is not a lipoprotein receptor protein-1, but LDL lipoprotein receptor-related protein-1).

Response: We have corrected the terms. Thank you.

Comment 8. Why microglia medium was used in a series of experiments (supp. fig. 1)?

Response: We believe that glial network in a neurovascular unit should share culture media in *in vitro* culture systems ultimately. We also note that previous studies have reported that culture medium for astrocytes and microglia can be shared; especially, astrocytes grow well in both astrocyte medium and microglia medium (Saura, 2007). Moreover, we will ultimately develop a better neurovascular unit system including the current BBB model, in which microglia will be co-cultured.

Comment 9. The obtained value of TEER (200 for BBB model) seems to be rather low comparing with other microfluidic BBB models (i.e. Y.I. Wang et al., 2017). Also, it is not clear whether shear stress was able to increase TEER.

Response: TEER values for in vitro BBB models with immortalized or primary endothelial cells are in the range of 100~500 $\Omega \cdot \text{cm}^2$, which varies on the culture condition (e.g., time, flow, 2D vs. 3D) (DeStefano et al., 2018; Odijk et al., 2015). Recent attempts made in the development of *in vitro* BBB models as pointed by the reviewer include iPSC-derived endothelial cells, which show TEER values over 1000 $\Omega \cdot \text{cm}^2$ (Wang et al., 2017). Higher TEER values reported in very recent BBB models were thought to result from use of iPSC-derived endothelial cells (Park et al., 2019; Vatine et al., 2019).

In addition, for the reviewer's comment on TEER and shear stress relationship, we have conducted additional experiments to investigate the effect of shear stress (0, 0.4 and 4 dyne/cm²) on TEER values and added the results in the revised manuscript (Result) as also shown in our Response to Comment 6 above.

Added text:

On page 10, line 184- A physiological level of shear stress was responsible for inducing endothelial function with the barrier tightness (**Fig. 2I**), efflux transporter protein expression (**Supplementary Fig. 5**), and endothelial nitric oxide synthase (eNOS) phosphorylation (**Supplementary Fig. 6a and 6b**).

Comment 10. Fig.1i contains data obtained with MTS assay. Authors should specify in the text and in the figure legend what parameter they actually measured: viability or metabolic activity.

Response: We measured the metabolic activity of cells to provide optimal culture microenvironments for all three types of cells. We now clearly stated that in the revised manuscript (Result, Methods, and the Figure legend).

Added text:

On page 6, line 122 - With morphological and metabolic activity assays (see Methods for details)

On page 9, line 150 (Figure caption for Fig. 1i) – i, Cell metabolic activities assessed by a (3-(4,5-dimethylthiazol-2-yl)-5-(3-carboxymethoxyphenyl)-2-(4-sulfophenyl)-2H-tetrazolium) (MTS) assay

On page 26, line 454 - To optimize cell culture condition, cell metabolic activity viabilities in different medium condition were quantified by measuring cell metabolic activities using a (3-(4,5-dimethylthiazol-2-yl)-5-(3-carboxymethoxyphenyl)-2-(4-sulfophenyl)-2H-tetrazolium) (MTS) assay (n = 6), which measures the formazan product by cell metabolic activity.

REFERENCES

- Adriani, G., Ma, D., Pavesi, A., Kamm, R.D., and Goh, E.L. (2017). A 3D neurovascular microfluidic model consisting of neurons, astrocytes and cerebral endothelial cells as a blood-brain barrier. *Lab Chip* 17, 448-459.
- Amiry-Moghaddam, M., Williamson, A., Palomba, M., Eid, T., de Lanerolle, N.C., Nagelhus, E.A., Adams, M.E., Froehner, S.C., Agre, P., and Ottersen, O.P. (2003). Delayed K⁺ clearance associated with aquaporin-4 mislocalization: phenotypic defects in brains of alpha-syntrophin-null mice. *Proc Natl Acad Sci U S A* 100, 13615-13620.
- Bang, S., Lee, S.R., Ko, J., Son, K., Tahk, D., Ahn, J., Im, C., and Jeon, N.L. (2017). A Low Permeability Microfluidic Blood-Brain Barrier Platform with Direct Contact between Perfusable Vascular Network and Astrocytes. *Sci Rep* 7, 8083.
- Booth, R., and Kim, H. (2012). Characterization of a microfluidic in vitro model of the blood-brain barrier (mu BBB). *Lab on a Chip* 12, 1784-1792.
- Camassa, L.M.A., Lunde, L.K., Hoddevik, E.H., Stensland, M., Boldt, H.B., De Souza, G.A., Ottersen, O.P., and Amiry-Moghaddam, M. (2015). Mechanisms underlying AQP4 accumulation in astrocyte endfeet. *Glia* 63, 2073-2091.
- Campisi, M., Shin, Y., Osaki, T., Hajal, C., Chiono, V., and Kamm, R.D. (2018). 3D self-organized microvascular model of the human blood-brain barrier with endothelial cells, pericytes and astrocytes. *Biomaterials* 180, 117-129.
- Cho, H., Seo, J.H., Wong, K.H., Terasaki, Y., Park, J., Bong, K., Arai, K., Lo, E.H., and Irimia, D. (2015). Three-Dimensional Blood-Brain Barrier Model for in vitro Studies of Neurovascular Pathology. *Sci Rep* 5, 15222.
- Daneman, R., and Prat, A. (2015). The blood-brain barrier. *Cold Spring Harb Perspect Biol* 7, a020412.
- DeStefano, J.G., Jamieson, J.J., Linville, R.M., and Searson, P.C. (2018). Benchmarking in vitro tissue-engineered blood-brain barrier models. *Fluids Barriers CNS* 15, 32.
- Di Benedetto, B., Malik, V.A., Begum, S., Jablonowski, L., Gomez-Gonzalez, G.B., Neumann, I.D., and Rupprecht, R. (2016). Fluoxetine Requires the Endfeet Protein Aquaporin-4 to Enhance Plasticity of Astrocyte Processes. *Front Cell Neurosci* 10, 8.
- Eidsvaag, V.A., Enger, R., Hansson, H.A., Eide, P.K., and Nagelhus, E.A. (2017). Human and mouse cortical astrocytes differ in aquaporin-4 polarization toward microvessels. *Glia* 65, 964-973.
- Eigenmann, D.E., Xue, G., Kim, K.S., Moses, A.V., Hamburger, M., and Oufir, M. (2013). Comparative study of four immortalized human brain capillary endothelial cell lines, hCMEC/D3, hBMEC, TY10, and BB19, and optimization of culture conditions, for an in vitro blood-brain barrier model for drug permeability studies. *Fluids Barriers CNS* 10, 33.
- Fung, K.Y., Wang, C., Nyegaard, S., Heit, B., Fairn, G.D., and Lee, W.L. (2017). SR-BI Mediated Transcytosis of HDL in Brain Microvascular Endothelial Cells Is Independent of Caveolin, Clathrin, and PDZK1. *Front Physiol* 8, 841.
- Herland, A., van der Meer, A.D., FitzGerald, E.A., Park, T.E., Sleeboom, J.J., and Ingber, D.E. (2016). Distinct Contributions of Astrocytes and Pericytes to Neuroinflammation Identified in a 3D Human Blood-Brain Barrier on a Chip. *PLoS One* 11, e0150360.
- Hung, P.J., Lee, P.J., Sabounchi, P., Aghdam, N., Lin, R., and Lee, L.P. (2005). A novel high aspect ratio microfluidic design to provide a stable and uniform microenvironment for cell growth in a high throughput mammalian cell culture array. *Lab Chip* 5, 44-48.
- Jamieson, J.J., Linville, R.M., Ding, Y.Y., Gerecht, S., and Searson, P.C. (2019). Role of iPSC-derived pericytes on barrier function of iPSC-derived brain microvascular endothelial cells in 2D and 3D. *Fluids Barriers CNS* 16, 15.
- Jones, V.C., Atkinson-Dell, R., Verkhatsky, A., and Mohamet, L. (2017). Aberrant iPSC-derived human astrocytes in Alzheimer's disease. *Cell Death Dis* 8, e2696.
- Kim, Y., Joshi, S.D., Davidson, L.A., LeDuc, P.R., and Messner, W.C. (2011). Dynamic control of 3D chemical profiles with a single 2D microfluidic platform. *Lab Chip* 11, 2182-2188.
- Kim, Y., Lobatto, M.E., Kawahara, T., Lee Chung, B., Mieszawska, A.J., Sanchez-Gaytan, B.L., Fay, F., Senders, M.L., Calcagno, C., Becraft, J., et al. (2014). Probing nanoparticle translocation across the permeable endothelium in experimental atherosclerosis. *Proc Natl Acad Sci U S A* 111, 1078-1083.
- Lillis, A.P., Van Duyn, L.B., Murphy-Ullrich, J.E., and Strickland, D.K. (2008). LDL receptor-related protein 1: unique tissue-specific functions revealed by selective gene knockout studies. *Physiol Rev* 88, 887-918.

- Maoz, B.M., Herland, A., FitzGerald, E.A., Grevesse, T., Vidoudez, C., Pacheco, A.R., Sheehy, S.P., Park, T.E., Dauth, S., Mannix, R., *et al.* (2018). A linked organ-on-chip model of the human neurovascular unit reveals the metabolic coupling of endothelial and neuronal cells. *Nat Biotechnol* 36, 865-874.
- Nagelhus, E.A., and Ottersen, O.P. (2013). Physiological roles of aquaporin-4 in brain. *Physiol Rev* 93, 1543-1562.
- Neely, J.D., Amiry-Moghaddam, M., Ottersen, O.P., Froehner, S.C., Agre, P., and Adams, M.E. (2001). Syntrophin-dependent expression and localization of Aquaporin-4 water channel protein. *Proc Natl Acad Sci U S A* 98, 14108-14113.
- Nicchia, G.P., Frigeri, A., Liuzzi, G.M., Santacroce, M.P., Nico, B., Procino, G., Quondamatteo, F., Herken, R., Roncali, L., and Svelto, M. (2000). Aquaporin-4-containing astrocytes sustain a temperature- and mercury-insensitive swelling in vitro. *Glia* 31, 29-38.
- Nicchia, G.P., Frigeri, A., Liuzzi, G.M., and Svelto, M. (2003). Inhibition of aquaporin-4 expression in astrocytes by RNAi determines alteration in cell morphology, growth, and water transport and induces changes in ischemia-related genes. *FASEB J* 17, 1508-1510.
- Odijk, M., van der Meer, A.D., Levner, D., Kim, H.J., van der Helm, M.W., Segerink, L.I., Frimat, J.P., Hamilton, G.A., Ingber, D.E., and van den Berg, A. (2015). Measuring direct current trans-epithelial electrical resistance in organ-on-a-chip microsystems. *Lab Chip* 15, 745-752.
- Pardridge, W.M. (2002). Drug and gene targeting to the brain with molecular Trojan horses. *Nat Rev Drug Discov* 1, 131-139.
- Park, J., Wetzel, I., Marriott, I., Dreau, D., D'Avanzo, C., Kim, D.Y., Tanzi, R.E., and Cho, H. (2018). A 3D human triculture system modeling neurodegeneration and neuroinflammation in Alzheimer's disease. *Nat Neurosci* 21, 941-951.
- Park, T.E., Mustafaoglu, N., Herland, A., Hasselkus, R., Mannix, R., FitzGerald, E.A., Prantil-Baun, R., Watters, A., Henry, O., Benz, M., *et al.* (2019). Hypoxia-enhanced Blood-Brain Barrier Chip recapitulates human barrier function and shuttling of drugs and antibodies. *Nat Commun* 10, 2621.
- Placone, A.L., McGuiggan, P.M., Bergles, D.E., Guerrero-Cazares, H., Quinones-Hinojosa, A., and Searson, P.C. (2015). Human astrocytes develop physiological morphology and remain quiescent in a novel 3D matrix. *Biomaterials* 42, 134-143.
- Saadoun, S., Papadopoulos, M.C., Watanabe, H., Yan, D., Manley, G.T., and Verkman, A.S. (2005). Involvement of aquaporin-4 in astroglial cell migration and glial scar formation. *J Cell Sci* 118, 5691-5698.
- Saura, J. (2007). Microglial cells in astroglial cultures: a cautionary note. *J Neuroinflammation* 4, 26.
- Steiner, J., Bernstein, H.G., Bielau, H., Berndt, A., Brisch, R., Mawrin, C., Keilhoff, G., and Bogerts, B. (2007). Evidence for a wide extra-astrocytic distribution of S100B in human brain. *BMC Neurosci* 8, 2.
- Storck, S.E., Meister, S., Nahrath, J., Meissner, J.N., Schubert, N., Di Spiezio, A., Baches, S., Vandenbroucke, R.E., Bouter, Y., Prikulis, I., *et al.* (2016). Endothelial LRP1 transports amyloid-beta(1-42) across the blood-brain barrier. *J Clin Invest* 126, 123-136.
- Thorsen, T., Maerkl, S.J., and Quake, S.R. (2002). Microfluidic large-scale integration. *Science* 298, 580-584.
- Toth, M.J., Kawahara, T., and Kim, Y. (2018). High-precision microfluidic pressure control through modulation of dual fluidic resistances. *International Journal of Dynamics and Control* 6, 1175-1182.
- Toth, M.J., Kim, T., and Kim, Y. (2017). Robust manufacturing of lipid-polymer nanoparticles through feedback control of parallelized swirling microvortices. *Lab Chip* 17, 2805-2813.
- Van Eck, M., Twisk, J., Hoekstra, M., Van Rij, B.T., Van der Lans, C.A., Bos, I.S., Kruijt, J.K., Kuipers, F., and Van Berkel, T.J. (2003). Differential effects of scavenger receptor BI deficiency on lipid metabolism in cells of the arterial wall and in the liver. *J Biol Chem* 278, 23699-23705.
- Vatine, G.D., Barrile, R., Workman, M.J., Sances, S., Barriga, B.K., Rahnama, M., Barthakur, S., Kasendra, M., Lucchesi, C., Kerns, J., *et al.* (2019). Human iPSC-Derived Blood-Brain Barrier Chips Enable Disease Modeling and Personalized Medicine Applications. *Cell Stem Cell* 24, 995-1005 e1006.
- Wang, J.D., Khafagy el, S., Khanafer, K., Takayama, S., and ElSayed, M.E. (2016). Organization of Endothelial Cells, Pericytes, and Astrocytes into a 3D Microfluidic in Vitro Model of the Blood-Brain Barrier. *Mol Pharm* 13, 895-906.

- Wang, Y.I., Abaci, H.E., and Shuler, M.L. (2017). Microfluidic blood-brain barrier model provides in vivo-like barrier properties for drug permeability screening. *Biotechnol Bioeng* 114, 184-194.
- White, A.K., VanInsberghe, M., Petriv, O.I., Hamidi, M., Sikorski, D., Marra, M.A., Piret, J., Aparicio, S., and Hansen, C.L. (2011). High-throughput microfluidic single-cell RT-qPCR. *Proc Natl Acad Sci U S A* 108, 13999-14004.
- Yi, Y., Park, J., Lim, J., Lee, C.J., and Lee, S.H. (2015). Central Nervous System and its Disease Models on a Chip. *Trends Biotechnol* 33, 762-776.
- Yuan, W., Lv, Y., Zeng, M., and Fu, B.M. (2009). Non-invasive measurement of solute permeability in cerebral microvessels of the rat. *Microvasc Res* 77, 166-173.

Reviewers' comments:

Reviewer #1 (Remarks to the Author):

The authors took a critical look to the comments of the reviewers. However, there are still some remaining points to be clarified, before it is ready for publication

* the authors now mention shear stresses and flows, based on the suggestions of R1. However, things are still unknown for the reader, making it difficult to (re)do the experiments. So, what are the channel dimensions of the chip?

* comment 4 of R1. I do not think figure 1 h make any sense at this moment. It does not take into account the effect of the endothelial layer (and if everything is going well, this one is very tight). Therefore I think it is confusing, since it does not take into account one of the most important characteristics of the barrier. Additionally the take up of the astrocytes is not taken into account. So the situation of t-60 minutes is quite okayish, since the nutrients are everywhere. I think the model is not showing what the authors want to show and therefore not suited to do any statements about it.

* the authors claim that they developed a high throughput co-culture protocol. This is not valid for the device shown here. Yes, the authors are right in the response that they can make it like that, but that is not the case here. Please remove high throughput.

* the way the TEER measurements are done is still prone to make errors. See for an explanation the articles of Odijk et al. in Lab Chip 2015 15(3) or van der Helm et al. Lab Chip 2019. Here you can see that a long membrane in a microfluidic channel will not lead to an even potential across the membrane. This means that different parts of the membrane will be more dominant for the measured resistance than others.

Also it seems that the resistance of the channel is not subtracted from the value. This needs to be done otherwise the length of the channel plays a role as well.

* the authors gave a reasoning why they think high reproducibility is fair. Although I can follow this, I think by making such a statement, you should also mention/quantify what you mean. The authors are not doing this, and this should be done (or high reproducibility should be taken out).

Reviewer #2 (Remarks to the Author):

I agree with all the authors' responses and corrections made according to my comments.

Responses to Reviewers

Reviewer #1

Comment 1. The authors now mention shear stresses and flows, based on the suggestions of R1. However, things are still unknown for the reader, making it difficult to (re)do the experiments. So, what are the channel dimensions of the chip?

Response: The widths of the upper, lower center, and lower side channels are 400 μm , 300 μm , and 200 μm , respectively. The height of all channels is 100 μm . We have now stated the channel dimensions in our manuscript.

Added (Revised) text:

On page 24, line 423 – The microfluidic device was designed to have the widths of the upper, lower center, and lower side channels of 400 μm , 300 μm , and 200 μm respectively. The height of all channels is 100 μm .

Comment 2. Comment 4 of R1. I do not think figure 1 h make any sense at this moment. It does not take into account the effect of the endothelial layer (and if everything is going well, this one is very tight). Therefore I think it is confusing, since it does not take into account one of the most important characteristics of the barrier. Additionally the take up of the astrocytes is not taken into account. So the situation of t-60 minutes is quite okayish, since the nutrients are everywhere. I think the model is not showing what the authors want to show and therefore not suited to do any statements about it.

Response: We thank the reviewer for the constructive comment. We wanted to leverage the computational model to design a microfluidic device that provide enough perfusion to cells in 3D without consideration of cellular components and their roles in the BBB model. We agree with the reviewer that the current presentation of Figure 1h makes it confusing, as the computational model does not include the characteristics of the barrier and cellular function. Therefore, we removed Figure 1h from the main figure while showing the computational model in Supplementary Figure 1 with captions clearly describing the simple channel design purpose and conditions without cells. We will also be open to more discussion if the reviewer thinks it should be out of the manuscript.

Added (Revised) text:

On page 6, line 116 - The device is designed to have diffusive transport of culture medium components into the hydrogel channel with media refreshment in the upper and the two side channels (**Supplementary Fig. 1**).

Comment 3. The authors claim that they developed a high throughput co-culture protocol. This is not valid for the device shown here. Yes, the authors are right in the response that they can make it like that, but that is not the case here. Please remove high throughput.

Response: We understand the reviewer's point, so we removed the "high-throughput" from the manuscript.

Comment 4. The way the TEER measurements are done is still prone to make errors. See for an explanation the articles of Odijk et al. in Lab Chip 2015 15(3) or van der Helm et al. Lab Chip 2019. Here you can see that a long membrane in a microfluidic channel will not lead to an even potential across the membrane. This means that different parts of the membrane will be more dominant for the measured resistance than others.

Also it seems that the resistance of the channel is not subtracted from the value. This needs to be done otherwise the length of the channel plays a role as well.

Response: As the reviewer suggested, we subtracted the resistance of the channel from the values. We understand the reviewer's concern that the resistance will not be homogeneous over the entire long membrane. To avoid the measurement errors by variations in the electrode placement (channel length), the electrodes were positioned in the same position in the device throughout the experiments. Moreover, the EVOM2 generates a constant current with the frequency of 12.5 Hz, which shows low variation of potential distribution along the long channel compared to the higher frequencies, as shown in van der Helm et al. Lab Chip 2019.

Added text:

On page 30, line 545 - To calculate TEER, the measurements from the chips in the absence of the cells were subtracted from the resistance of each device, and then the values were multiplied by the surface area of endothelial monolayer overlapping with the lower channel (0.015 cm²).

Comment 5. The authors gave a reasoning why they think high reproducibility is fair. Although I can follow this, I think by making such a statement, you should also mention/quantify what you mean. The authors are not doing this, and this should be done (or high reproducibility should be taken out).

Response: Thank you for the comments. We admit that the term "high" is subjective and relative, so we deleted the term from the manuscript. Nevertheless, to clarify our experiments, we have stated in the revised manuscript that all images in the manuscript are representative of at least three biological and at least three technical replicates. The high reproducibility of our system can be shown with the experimental results (TEER, permeability, NP sampling) that have been collected from at least 3 repetitive experiments and the results (AQP4 polarization and FACS analysis) from 2 repetitive experiment. Moreover, at least 2 replicates of each group were used per experiment. We now stated the number of experimental repetition and the number of biological and technical replicates in each figure legend and Method (Statistical analyses).

Added text:

On page 8, line 156 – All images are representative ones from at least five biological and three technical replicates.

On page 12, line 212 – All images are representative ones from at least five biological and three technical replicates.

On page 16, line 279 – All images are representative ones from at least five biological and three technical replicates.

On page 20, line 341 – All images are representative ones from at least three biological and three technical replicates.

On page 35, line 634 – All chip experiments were reproduced for at least two times to confirm data reliability. Per experiment, at least two biological and technical replicates were used.

Reviewer #2

General Comments:

I agree with all the authors' responses and corrections made according to my comments.

Response: We thank for the reviewer's input which greatly improved the quality of our manuscript.

Reviewers' comments:

Reviewer #1 (Remarks to the Author):

Thanks for considering all the suggestions of the reviewers.

#1; thanks for adding the dimensions

#2: Your results suggest that the diffusion is enough to give enough nutrients to the cells. I think it is a nice first guess and therefore I like it, but the readers should be aware that the cells are not taken into account and that one should be taking care with the results. However since the cells are vialbe after some time, it seems that it is okay. But in my opinion the authors stress the model too much.

#3. Thanks

4 However I am still concerned about the TEER values and the way the measurements is done. Also at 12.5Hz, you will have an uneven distribution of the potential accross the membrane, where different parts will be more highlighted than others. By placing the electrodes on the same position (if that is possible) this will not be solved and that is my point.

#5. thanks

Reviewer #3 (Remarks to the Author):

For the TEER measurements:

1. The methods for measuring TEER p. 30, l. 538 are not detailed enough to follow the exact procedure: the method employs custom Ag/AgCl electrodes linked to a commercially available EVOM2 Volt/Ohmmeter, but it is not clear how these electrodes were linked to the commercial apparatus. This would particularly be important because the when linked to the commercial electrodes of WPI, the EVOM2 depends on a 4-point measurement electrode configuration (two driving electrodes, two sensor electrodes). Further details/references should be provided.
2. The inhomogeneous current distribution over the length of the measured membrane area is something that will always manifest itself when measuring electrical resistance of cell monolayers in microfluidic chips, and without correction, this phenomenon will always lead one to overestimate their TEER values (see Odijk et al. Lab Chip 2015, Van der Helm et al. Lab Chip 2019, Yeste et al. Journal of Physics D: Applied Physics 2016). The exact magnitude of the overestimation is dependent on channel dimensions (both cross-section and membrane area), as well as the electrical resistance of the cell layer.

I would suggest that the authors address this issue by (1) acknowledging that this effect indeed exists and will also apply to their measurements because they were performed in a microfluidic chip, and by either (2) attempting to get an estimate of the maximum percentage error that they have made when calculating their TEER, or (3) reporting only raw resistance data with a note that the collected resistance data don't scale linearly with the actual cell layer resistance/TEER.

For point (2), if the channel dimensions, electrode configurations and approximate electrical resistance of the cell layer are comparable to one of the conditions reported in any of the aforementioned papers, then these papers could be used to get an estimate of the maximum percentage error the authors are making by not correcting for inhomogeneous current distribution over the membrane. If not, the authors will have to calculate the corrected TEER values themselves (by using the Kirchoff matrix approach in the Odijk and Van der Helm papers, or by applying the analytical model given in the Suppl. Inf. of the Odijk paper, or by applying a COMSOL simulation like in the Yeste paper).

Responses to Reviewers

Reviewer #1

Comment 2. Your results suggest that the diffusion is enough to give enough nutrients to the cells. I think it is a nice first guess and therefore I like it, but the readers should be aware that the cells are not taken into account and that one should be taking care with the results. However since the cells are viable after some time, it seems that it is okay. But in my opinion the authors stress the model too much.

Response: In the figure caption (**Supplementary Figure 1**), we have stated that the effect of cells cultured in microfluidic channels are not considered in this computational model. To destress the model, as suggested, we have shown only two different time points as below.

Revised text (and figure):

On page 41, Supplementary Figure 1

(Cells are not considered in the computational modeling)

Supplementary Figure 1. Computational fluid dynamic simulation of diffusion through the microfluidic channels in the device. This model has not considered the effect of cells cultured in the device but has simply focused on diffusive transport into the hydrogel channel (lower center channel) in this hybrid channel configuration.

Comment 4. However I am still concerned about the TEER values and the way the measurements is done. Also at 12.5Hz, you will have an uneven distribution of the potential across the membrane, where different parts will be more highlighted than others. By placing the electrodes on the same position (if that is possible) this will not be solved and that is my point.

Response: We admit that the TEER measurement could be affected by the uneven potential distribution, which is reported in previous studies for comparison with Transwell models and for more accurate TEER measurement using impedance (Odijk et al., 2015; van der Helm et al., 2019; Yeste et al., 2016). However, as pointed through the manuscript, more important point in this study is to replicate more physiologically relevant BBB with astrocytes cultured in 3D hydrogel, where electrodes could not be coated along the bottom hydrogel channel as reported and discussed in previous studies (Arik et al., 2018; Herland et al., 2016). Therefore, in addition to our evaluation of the barrier function with molecular permeability (permeability coefficient), we attempted to measure the TEER values from the electrode wires inserted in the end of the inlet and

outlet of the channels, as shown in previous studies (Kim et al., 2014; Partyka et al., 2017; Xu et al., 2016), in order to cross-validate the barrier function and ensure that our BBB model shows higher integrity than the EC only model constructed in the same device structure (not comparing with Transwell or other models). Although the TEER values may not indicate the accurate values of the cell layer resistance due to the uneven distribution, we demonstrated that the TEER value of the BBB is higher than the TEER of the EC only culture model with 12 biological replicates (12 chips).

Demonstrating that the permeability coefficients are comparable to those measured *in vivo* (Yuan et al., 2009), our approach using two assays to verify the difference between the EC only and BBB models is convincing. In order to avoid confusion about the TEER measurements, we have stated that the TEER values can be affected by the uneven distribution of the potential across the membrane and that our TEER values were used to compare the barrier integrity between the EC only and the BBB models in the device.

Added text:

On page 21, In addition to our evaluation of the barrier function with molecular permeability (permeability coefficient analysis), we measured the TEER values from the electrode wires inserted in the end of the inlet and outlet of channels, as shown in previous studies (Kim et al., 2014; Partyka et al., 2017; Xu et al., 2016), in order to cross-validate the barrier function difference between our models in the same device (not comparing them with Transwell or other models). We note that 3D hydrogel in the lower channel prevents electrodes from being coated along the bottom hydrogel channel as reported in recent studies (Arik et al., 2018; Herland et al., 2016) and that the TEER values measured in our current study can be affected by the uneven distribution of the potential across the membrane and may not indicate the accurate values of the cell layer resistance as reported in recent studies (Odijk et al., 2015; van der Helm et al., 2019; Yeste et al., 2016). The TEER values measured in this study thus were simply used to compare the barrier integrity between the EC only and the BBB models in the same device for the purpose of cross-validation of permeability coefficient analysis. For more information, we have included the raw resistance data in **Supplementary Table 2**.

On page 53, Supplementary Table 2

#	EC only	BBB (4 dyne/cm ²)	BBB (No shear)	BBB (0.4 dyne/cm ²)
1	7320	10373	7485	8960
2	7127	9712	6309	6974
3	9407	10399	7030	9175
4	9515	9782	7703	9372
5	5690	9412	6767	
6	7907	9217		
7	9059	9441		
8	9522	9476		
9	9659	9734		
10	5835	9107		
11	9583	9311		
12		9528		

unit: Ω

Supplementary Table 2. Raw resistance data measured using EVOM2. Raw resistance data of the endothelial monolayer only model (EC only) and the BBB model (under different shear stress levels). The data may not be linearly correlated to the actual cell layer resistance values due to the inhomogeneous potential distribution over the channel length.

Reviewer #3

For the TEER measurements:

Comment 1. The methods for measuring TEER p. 30, l. 538 are not detailed enough to follow the exact procedure: the method employs custom Ag/AgCl electrodes linked to a commercially available EVOM2 Volt/Ohmmeter, but it is not clear how these electrodes were linked to the commercial apparatus. This would particularly be important because the when linked to the commercial electrodes of WPI, the EVOM2 depends on a 4-point measurement electrode configuration (two driving electrodes, two sensor electrodes). Further details/references should be provided.

Response: We have provided further details, references, and also a schematic (**Supplementary Figure 11**) to better describe our TEER measurement. TEER was measured following a protocol as previously reported (Kim et al., 2014), where a custom electrical connector using Rj11 plug and Ag, Ag/AgCl electrode wires (381 μm in diameter, A-M Systems) were designed to insert into the microfluidic channels. Moreover, the electrode wires were placed in a tygon tubing filled with culture medium to reduce the possible background resistance and error as previously reported (Kim et al., 2014).

Added text (and figure):

On page 29, The TEER of the endothelial monolayer formed in the device was measured using a custom electrode adaptor (Kim et al., 2014; Sei et al., 2017) made with Rj11 plug and Ag, Ag/AgCl electrode wires (381 μm in diameter and 3 cm in length, A-M Systems, Sequim, WA) connected to EVOM2 volt-ohmmeter (World Precision Instruments, Sarasota, FL) which generates a constant 10 μA of AC current at 12.5 Hz while measuring resistance. To reduce background resistance and error, the electrode wires were placed in a tubing (1/32"ID x 3/32"OD, Cole-Parmer, Vernon Hills, IL, USA) filled with culture medium and inserted into the channels (**Supplementary Fig. 11**). After 1 min of stabilization, 5 multiple readings were averaged for each device. To calculate TEER, the measurements from the chips in the absence of the cells were subtracted from the resistance of each device, and then the values were multiplied by the surface area of endothelial monolayer overlapping with the lower channel (0.015 cm^2).

On page 52, Supplementary Figure 11 (see the next page)

[redacted]

Supplementary Figure 11. Transendothelial electrical resistance (TEER) measurement. TEER was measured using a commercially available volt-ohmmeter (EVOM2) with a custom electrode adaptor made with Rj11 plug and Ag, Ag/AgCl electrode wires. The 3 cm electrode wires were placed in a tygon tubing filled with culture medium to reduce the possible background resistance and error as previously reported (Kim et al., 2014). Our custom electrode adaptor has four ports connected to the four Rj11 leads (two ports are connected to Ag wires to pass current and the other two ports are connected to Ag/AgCl wires to detect voltage). Each of the Ag and Ag/AgCl wire is inserted into the upper (indicated by blue circles) and the lower side channel (indicated by red circles).

Comment 2. The inhomogeneous current distribution over the length of the measured membrane area is something that will always manifest itself when measuring electrical resistance of cell monolayers in microfluidic chips, and without correction, this phenomenon will always lead one to overestimate their TEER values (see Odijk et al. Lab Chip 2015, Van der Helm et al. Lab Chip 2019, Yeste et al. Journal of Physics D: Applied Physics 2016). The exact magnitude of the overestimation is dependent on channel dimensions (both cross-section and membrane area), as well as the electrical resistance of the cell layer.

I would suggest that the authors address this issue by (1) acknowledging that this effect indeed exists and will also apply to their measurements because they were performed in a microfluidic chip, and by either (2) attempting to get an estimate of the maximum percentage error that they

have made when calculating their TEER, or (3) reporting only raw resistance data with a note that the collected resistance data don't scale linearly with the actual cell layer resistance/TEER.

For point (2), if the channel dimensions, electrode configurations and approximate electrical resistance of the cell layer are comparable to one of the conditions reported in any of the aforementioned papers, then these papers could be used to get an estimate of the maximum percentage error the authors are making by not correcting for inhomogeneous current distribution over the membrane. If not, the authors will have to calculate the corrected TEER values themselves (by using the Kirchoff matrix approach in the Odijk and Van der Helm papers, or by applying the analytical model given in the Suppl. Inf. of the Odijk paper, or by applying a COMSOL simulation like in the Yeste paper).

Response: We thank the reviewer for the constructive comment. We admit that the TEER measurement could be affected by the uneven potential distribution, which is reported in previous studies for comparison with Transwell models and for more accurate TEER measurement using impedance (Odijk et al., 2015; van der Helm et al., 2019; Yeste et al., 2016). However, as pointed through the manuscript, more important point in this study is to replicate more physiologically relevant BBB with astrocytes cultured in 3D hydrogel, where electrodes could not be coated along the bottom hydrogel channel as reported and discussed in previous studies (Arik et al., 2018; Herland et al., 2016). Therefore, in addition to our evaluation of the barrier function with molecular permeability (permeability coefficient), we attempted to measure the TEER values from the electrode wires inserted in the end of the inlet and outlet of the channels, as shown in previous studies (Kim et al., 2014; Partyka et al., 2017; Xu et al., 2016), in order to cross-validate the barrier function and ensure that our BBB model shows higher integrity than the EC only model constructed in the same device structure (not comparing with Transwell or other models). Although the TEER values may not indicate the accurate values of the cell layer resistance due to the uneven distribution, we demonstrated that the TEER value of the BBB is higher than the TEER of the EC only culture model with 12 biological replicates (12 chips). Demonstrating that the permeability coefficients are comparable to those measured *in vivo* (Yuan et al., 2009), our approach using two assays to verify the difference between the EC only and BBB models is convincing. In order to avoid confusion about the TEER measurements, we have stated in our manuscript that the TEER values can be affected by the uneven distribution of the potential across the membrane, and that our TEER values were used to compare the barrier integrity between the EC only and the BBB models in the same device.

We have addressed the three points as below:

- 1) We have stated that the TEER values depend on the channel dimensions in the revised manuscript (Discussion).
- 2) Our device has three parallel bottom channels, with the center channel filled with Matrigel to construct a 3D microenvironment for astrocytes, and thus, we cannot estimate the corrected TEER using the results obtained from the previous studies which have all channels filled with culture medium. Moreover, due to the complex geometry of our device that has two vertical (top and bottom) and three parallel bottom channels compartmentalized with a series of micropillars, the corrected TEER values could not be assessed using the model equations. Instead, we have provided raw resistance data as suggested in the reviewer's point (3).

- 3) We included raw resistance data with a note that the data don't scale linearly with the actual cell layer resistance in the revised manuscript (Supplementary Table 2).

Added text (and table):

On page 21, In addition to our evaluation of the barrier function with molecular permeability (permeability coefficient analysis), we measured the TEER values from the electrode wires inserted in the end of the inlet and outlet channels, as shown in previous studies (Kim et al., 2014; Partyka et al., 2017; Xu et al., 2016), in order to cross-validate the barrier function difference between our models in the same device (not comparing them with Transwell or other models). We note that 3D hydrogel in the lower channel prevents electrodes from being coated along the bottom hydrogel channel as reported in recent studies (Arik et al., 2018; Herland et al., 2016) and that the TEER values measured in our current study can be affected by the uneven distribution of the potential across the membrane and may not indicate the accurate values of the cell layer resistance as reported in recent studies (Odijk et al., 2015; van der Helm et al., 2019; Yeste et al., 2016). The TEER values measured in this study thus were simply used to compare the barrier integrity between the EC only and the BBB models in the same device for the purpose of cross-validation of permeability coefficient analysis. For more information, we have included the raw resistance data in **Supplementary Table 2**.

On page 53, Supplementary Table 2

#	EC only	BBB (4 dyne/cm ²)	BBB (No shear)	BBB (0.4 dyne/cm ²)
1	7320	10373	7485	8960
2	7127	9712	6309	6974
3	9407	10399	7030	9175
4	9515	9782	7703	9372
5	5690	9412	6767	
6	7907	9217		
7	9059	9441		
8	9522	9476		
9	9659	9734		
10	5835	9107		
11	9583	9311		
12		9528		

unit: Ω

Supplementary Table 2. Raw resistance data measured using EVOM2. Raw resistance data of the endothelial monolayer only model (EC only) and the BBB model (under different shear stress levels). The data may not be linearly correlated to the actual cell layer resistance values due to the inhomogeneous potential distribution over the channel length.

References

- Arik, Y.B., van der Helm, M.W., Odijk, M., Segerink, L.I., Passier, R., van den Berg, A., and van der Meer, A.D. (2018). Barriers-on-chips: Measurement of barrier function of tissues in organs-on-chips. *Biomicrofluidics* 12, 042218.
- Herland, A., van der Meer, A.D., FitzGerald, E.A., Park, T.E., Sleeboom, J.J., and Ingber, D.E. (2016). Distinct Contributions of Astrocytes and Pericytes to Neuroinflammation Identified in a 3D Human Blood-Brain Barrier on a Chip. *PLoS One* 11, e0150360.
- Kim, Y., Lobatto, M.E., Kawahara, T., Lee Chung, B., Mieszawska, A.J., Sanchez-Gaytan, B.L., Fay, F., Senders, M.L., Calcagno, C., Becraft, J., *et al.* (2014). Probing nanoparticle translocation across the permeable endothelium in experimental atherosclerosis. *Proc Natl Acad Sci U S A* 111, 1078-1083.

- Odijk, M., van der Meer, A.D., Levner, D., Kim, H.J., van der Helm, M.W., Segerink, L.I., Frimat, J.P., Hamilton, G.A., Ingber, D.E., and van den Berg, A. (2015). Measuring direct current trans-epithelial electrical resistance in organ-on-a-chip microsystems. *Lab Chip* 15, 745-752.
- Partyka, P.P., Godsey, G.A., Galie, J.R., Kosciuk, M.C., Acharya, N.K., Nagele, R.G., and Galie, P.A. (2017). Mechanical stress regulates transport in a compliant 3D model of the blood-brain barrier. *Biomaterials* 115, 30-39.
- Sei, Y.J., Ahn, S.I., Virtue, T., Kim, T., and Kim, Y. (2017). Detection of frequency-dependent endothelial response to oscillatory shear stress using a microfluidic transcellular monitor. *Sci Rep* 7, 10019.
- van der Helm, M.W., Henry, O.Y.F., Bein, A., Hamkins-Indik, T., Crouce, M.J., Leineweber, W.D., Odijk, M., van der Meer, A.D., Eijkel, J.C.T., Ingber, D.E., *et al.* (2019). Non-invasive sensing of transepithelial barrier function and tissue differentiation in organs-on-chips using impedance spectroscopy. *Lab Chip* 19, 452-463.
- Xu, H., Li, Z., Yu, Y., Sizdahkhani, S., Ho, W.S., Yin, F., Wang, L., Zhu, G., Zhang, M., Jiang, L., *et al.* (2016). A dynamic in vivo-like organotypic blood-brain barrier model to probe metastatic brain tumors. *Sci Rep* 6, 36670.
- Yeste, J., Illa, X., Gutierrez, C., Sole, M., Guimera, A., and Villa, R. (2016). Geometric correction factor for transepithelial electrical resistance measurements in transwell and microfluidic cell cultures. *J Phys D Appl Phys* 49.
- Yuan, W., Lv, Y., Zeng, M., and Fu, B.M. (2009). Non-invasive measurement of solute permeability in cerebral microvessels of the rat. *Microvasc Res* 77, 166-173.

REVIEWERS' COMMENTS:

Reviewer #3 (Remarks to the Author):

All points have been addressed satisfactorily. I would like to thank the authors for their detailed response and relevant modification to their manuscript.